# Reading Between the Tokens: Improving Preference Predictions through Mechanistic Forecasting

**Sarah Ball** [* 1 2]   **Simeon Allmendinger** [* 3 4]   **Frauke Kreuter** [1 2 5]   **Niklas Kühl** [3 4]

## Abstract

Large language models are increasingly used to predict human preferences in both scientific and business endeavors, yet current approaches rely exclusively on analyzing model outputs without considering the underlying mechanisms. Using election forecasting as a test case, we introduce *mechanistic forecasting*, a method that demonstrates that probing internal model representations offers a fundamentally different—and sometimes more effective—approach to preference prediction. Examining over 24 million configurations across 7 models, 6 national elections, multiple persona attributes, and prompt variations, we systematically analyze how demographic and ideological information activates latent party-encoding components within the respective models. We find that leveraging this internal knowledge via mechanistic forecasting, opposed to solely relying on surface-level predictions, can improve prediction accuracy. The effects vary across demographic versus opinion-based attributes, political parties, national contexts, and models. Our findings demonstrate that the latent representational structure of LLMs contains systematic, exploitable information about human preferences, establishing a new path for using language models in social science prediction tasks.

## 1. Introduction

One particularly controversial application of AI is using LLMs for public opinion research and forecasting: instead

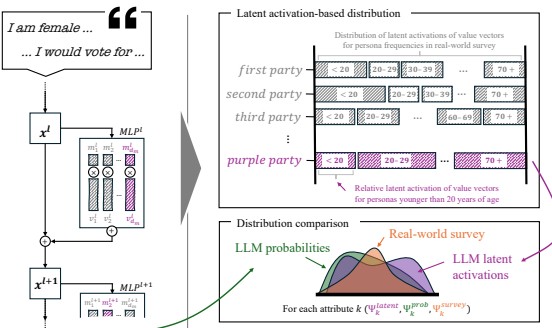

*Figure 1.* Overview of the proposed mechanistic forecasting method. Persona prompts are mapped to party-aligned latent value-vector activations inside the LLM. These activations are aggregated across personas into party-level preference distributions and compared against real-world survey outcomes.

of surveying humans, one prompts models with synthetic personas and aggregates their responses to approximate population-level preferences (Argyle et al., 2023; von der Heyde et al., 2024; Yu et al., 2024). This approach appears appealing because LLMs are trained on vast corpora containing countless expressions of political attitudes and social behavior. Yet, existing evidence on whether this works paints a mixed picture (Argyle et al., 2023; Kim & Lee, 2023). While persona-based LLM predictions can sometimes approximate average survey outcomes, they are often unstable to prompt phrasing, sensitive to alignment and instruction tuning, and uneven across countries, languages, and demographic groups (Bisbee et al., 2023; von der Heyde et al., 2024). These limitations raise a fundamental question: if LLMs encode knowledge about human preferences, why does so much of it fail to surface reliably in their outputs?

In this paper, we argue that the core bottleneck lies not in the absence of relevant information inside LLMs, but in how this information is elicited. Most existing approaches treat the model's final output distribution as the sole object of interest, evaluating performance entirely at the surface level (Bisbee et al., 2023; Brand et al., 2023; Jia et al., 2024). We instead reframe LLM-based social simulation as a problem of aggregating latent representations. Drawing on recent advances in mechanistic interpretability, we build on the insight that models often encode richer, more structured

---

[*]Equal contribution  [1]Department of Statistics, LMU Munich, Munich, Germany  [2]Munich Center for Machine Learning, Munich, Germany  [3]AI Responsibility Centre, University of Bayreuth, Bayreuth, Germany  [4]Fraunhofer Institute for Applied Information Technology, Germany  [5]JPSM, University of Maryland, College Park, Maryland, USA. Correspondence to: Sarah Ball <sarah.ball@stat.uni-muenchen.de>.

*Proceedings of the 43rd International Conference on Machine Learning*, Seoul, South Korea. PMLR 306, 2026. Copyright 2026 by the author(s).

knowledge internally than they reveal through generated answers—a phenomenon commonly referred to as latent or hidden knowledge (Gekhman et al., 2025; Orgad et al., 2025). From this perspective, failures of persona-based predicting may arise when output probabilities collapse or distort internal signals that, in fact, systematically reflect existing real life human preferences.

Our empirical results substantiate this view at scale. Analyzing a combinatorial space exceeding 24 million persona configurations across seven model families and six national election contexts, we show that aggregating internal activations tied to political parties frequently yields preference distributions closer to representative real-world survey data than those obtained from next-token probabilities alone. Crucially, these gains are not uniform: whether latent information helps depends on which attributes define the persona and how heterogeneous the corresponding population is. This observation motivates a central focus of our study: understanding *how* different categories of persona attributes influence forecasting performance.

**Contributions.** First, we introduce **mechanistic forecasting**, a method that identifies party-aligned value vectors in multi-layer perceptrons (MLPs) and aggregates their persona-induced activations into group-level preference distributions directly comparable to survey data. Second, across models, countries, and parties, we show that **LLMs encode substantial latent information about political preferences** not reliably expressed in output probabilities, and that **exploiting this latent signal often improves party-level forecasts** with respect to real-world survey data. Third, we demonstrate **systematic differences across persona attribute categories**: for many countries, demographic attributes (e.g., age, education, employment) are better captured by latent estimators than by output probabilities, while opinion-based attributes exhibit greater cross-national variation. Fourth, we identify **attribute-level entropy of output probability distributions as a simple gating criterion** for when mechanistic forecasting should be preferred over probability-based prediction[1].

## 2. Related Work

**Using LLMs as Substitutes for Humans.** The advent of LLMs has sparked significant interest regarding their potential to serve as substitutes for human respondents (Argyle et al., 2023). This question is especially relevant for survey researchers in the social sciences, who are investigating whether responses generated by LLMs can reliably resemble those provided by humans in surveys (Argyle et al., 2023; Bisbee et al., 2023; Dominguez-Olmedo et al., 2025; Park et al., 2024; Qu & Wang, 2024; von der Heyde

---

[1]Our code is available on GitHub.

et al., 2024; Wang et al., 2025). Similar inquiries have emerged in fields such as market research (Brand et al., 2023; Sarstedt et al., 2024), annotation tasks (Törnberg, 2023; Ziems et al., 2024), experiments in psychology and economics (Aher et al., 2023; Jia et al., 2024), and human-computer-interaction (Hämäläinen et al., 2023; Törnberg, 2023), among others. The findings from these investigations are mixed. Some studies suggest that LLMs can reasonably approximate the average outcomes of human surveys (Argyle et al., 2023; Bisbee et al., 2023; Hämäläinen et al., 2023; Törnberg, 2023; Brand et al., 2023; Jia et al., 2024), while others highlight significant limitations, particularly in their inability to accurately represent the opinions of diverse demographic groups (Santurkar et al., 2023; von der Heyde et al., 2024; Sarstedt et al., 2024; Qu & Wang, 2024; Dominguez-Olmedo et al., 2025). However, a common limitation across these studies is their focus on surface-level comparisons, e.g., matching LLM output to human survey responses, without delving into the mechanisms by which opinions are encoded and represented within the models' latent spaces. We address this gap by studying how personas are mapped to preferences, using latent information to improve preference predictions.

**Hidden Knowledge in LLMs.** AI models can generate incorrect information, including hallucinations (Huang et al., 2025; Zhang et al., 2025a). Research suggests that LLMs' internal representations encode knowledge about the correctness of generated answers or statements (Kadavath et al., 2022; Azaria & Mitchell, 2023; Chen et al., 2024), which has led to methods that leverage hidden states to detect and mitigate hallucinations (Azaria & Mitchell, 2023; Chen et al., 2024; Kossen et al., 2024; Sriramanan et al., 2024; Zhang et al., 2025b) or arithmetic errors (Sun et al., 2025). Gekhman et al. (2025) investigate whether LLMs have more "internal" than "external" knowledge by testing if internal functions rank answers more accurately than external ones. They find strong evidence for hidden knowledge: internal scoring methods outperform external approaches across three LLMs, with 40% average improvement. They use probing classifiers (logistic regression) trained on hidden states to predict answer correctness. Both Gekhman et al. (2025) and Orgad et al. (2025) demonstrate that hidden states can encode correct answers even when models generate incorrect responses. Our work extends this line of research in two key ways. First, while prior work focuses on binary factual correctness—a verifiable classification task—we examine preference prediction, where ground truth is inherently distributional and contextual rather than objectively determinable. Second, methodologically, we move beyond probing for answer correctness at the level of individual inputs: we identify MLP value vectors that encode party-specific representations and use Monte Carlo aggregation over persona-induced activations to estimate population-

level, multivariate voting distributions rather than binary outcomes. This shift from correctness detection to Monte Carlo estimation of distributional preferences represents a conceptually different application of latent states in LLMs.

## 3. Models and Data

**Model Selection.** We evaluate a set of base and instruction-tuned LLMs spanning multiple model families and parameter scales, all of which satisfy the white-box requirement necessary for analyzing latent representations. Specifically, we use Llama 3.1 models at 8B parameters in both base and instruction-tuned variants (MetaAI, 2024), Mistral 7B models in base and instruction-tuned form (Jiang et al., 2023), Gemma 2 models at 9B parameters with and without instruction tuning (Riviere et al., 2024), and the 14B-parameter Qwen 3 model (Yang et al., 2025). This selection covers a diverse set of architectures, training pipelines, and alignment strategies, enabling a systematic comparison of latent preference representations across model families.

**Real World Comparison.** In order to compare our model predictions to real data, we use representative cross-sectional election surveys from several countries: United States [2024] (American National Election Studies, 2025), United Kingdom [2024] (Fieldhouse et al., 2024), Canada [2019] (Stephenson et al., 2020), Germany [2021] (GESIS – Leibniz Institute for the Social Sciences, 2024), Netherlands [2021] (Sipma, 2021), and New Zealand [2020] (Vowles et al., 2022). These representative surveys capture insights about citizens' political attitudes, preferences, and voting behaviours. To obtain a representative comparison baseline, we weight the data with socio-demographic survey weights that align the distributions to the marginal distributions of the respective census data.

**Personas.** Constructing the persona prompts requires two key design choices: first, which variables to include for voting predictions, and second, how to embed those variables in a template. Concerning the variables, we build on established voting predictors from political science with empirically grounded attribute categories taken from representative election surveys, following prior work (von der Heyde et al., 2024). All persona attributes and their values are specified in a country-specific configuration file (cf. Table 1 in Appendix), covering socio-demographics (`age`, `gender`, `education`, `hhincome`, `employment`), ideological self-placement (`political_orientation`), and issue positions (`immigration`, `inequality`) as well as the `year_of_election`. Concerning the template, we use the following scheme: starting from the prompt structure of von der Heyde et al. (2024), two authors fluent in German hand-crafted 5 paraphrases (LLM-based rephrases were not satisfactory), then created a sentence-reordered version of each, yielding 10 German variants that encode

the same persona but differ in wording and sentence order. These were machine-translated into the remaining languages and verified by native speakers for semantic equivalence and idiomatic correctness. Exemplary prompt schemes are shown in Table 2 in the Appendix; all 10 variants per country are released in our code repository[1]. While building 10 prompts exceeds what comparable studies use (e.g. Argyle et al., 2023; Bisbee et al., 2023; von der Heyde et al., 2024), no principled criterion currently exists for determining how many variations are sufficient. We hence call for future work to develop a rigorous framework for designing prompt variations of personas.

**Probe Data.** In order to identify MLP value vectors that are related to specific parties, we need to train probes that capture what these parties represent. To do so, we manually compile data containing the political positions of parties from established voting-advice applications across countries. For Germany, we use the "Wahl-O-Mat" (Bundeszentrale für politische Bildung, 2025), an online questionnaire consisting of short political statements derived from party manifestos, to which parties provide a categorical stance and an explanatory comment. For the Netherlands, we analogously use data from the *StemjWizer* (ProDemos, 2026), and for all remaining countries (United Kingdom, United States, Canada, and New Zealand), we derive comparable party- or candidate-level position data from the *Vote Compass* (Vox Pop Labs, 2025). Across all countries, we manually collected and harmonized data to ensure semantic consistency of statements, responses, and explanations prior to probe training. The harmonized issue dataset and value-probe prompt templates are released in our code repository[1] (cf. Table 3 in Appendix for examples).

## 4. Introducing Mechanistic Forecasting

Our objective is to characterize how LLMs internally encode synthetic survey responses and how these internal representations relate to observed human preference distributions. Rather than relying on surface-level model outputs, we study the mapping from *persona descriptions* to *party representations* within the models' latent space. Building on recent work in mechanistic interpretability and probing (Elhage et al., 2021; Geva et al., 2022; Lee et al., 2024), our methodology proceeds in three steps: (i) we identify static MLP value vectors that promote party–related tokens, (ii) we quantify how persona prompts activate these vectors, and (iii) we aggregate these activations into multivariate distributions that are directly comparable to real-world survey data. Figure 1 provides an overview of this pipeline.

### 4.1. Technical Preliminaries

We consider an autoregressive transformer with $L$ layers and model dimension $d$. Let $x_i^l \in \mathbb{R}^d$ denote the residual stream

representation at token position $i$ after layer $l \in \{0, \dots, L\}$, with $x_i^0$ given by the token and positional embeddings. Each transformer layer consists of a multi-head self-attention (MHA) sublayer and a feed-forward MLP, both connected via residual connections. Ignoring bias terms and layer normalization for brevity, the residual update is given by (Elhage et al., 2021):

$$x_i^{l+1} = x_i^l + \mathrm{MLP}^l\left(x_i^l + \mathrm{MHA}^l(x_i^l)\right), \quad l = 0, \dots, L-1. \quad (1)$$

**MLP Decomposition.** Following Geva et al. (2022), we decompose each MLP into two linear maps with an element-wise nonlinearity in between. Let $d_{\mathrm{mlp}}$ denote the hidden width of the MLP. For an input vector $x^l \in \mathbb{R}^d$, the MLP computes

$$\mathrm{MLP}^l(x^l) = W_V^l\, f(W_K^l x^l), \quad (2)$$

where $W_K^l \in \mathbb{R}^{d_{\mathrm{mlp}} \times d}$, $W_V^l \in \mathbb{R}^{d \times d_{\mathrm{mlp}}}$, and $f(\cdot)$ is a pointwise activation function (e.g. GELU). Defining

$$m^l := f(W_K^l x^l) \in \mathbb{R}^{d_{\mathrm{mlp}}}, \quad (3)$$

the MLP output can be written as a linear combination of *value vectors*. Let $v_i^l \in \mathbb{R}^d$ denote the $i$-th column of $W_V^l$, and let $k_i^l \in \mathbb{R}^d$ denote the $i$-th row of $W_K^l$. Then

$$\mathrm{MLP}^l(x^l) = \sum_{i=1}^{d_{\mathrm{mlp}}} m_i^l\, v_i^l = \sum_{i=1}^{d_{\mathrm{mlp}}} f\left(\langle k_i^l, x^l \rangle\right) v_i^l. \quad (4)$$

**Interpretation as Sub-Updates.** Equation (4) shows that the MLP update decomposes into a sum of *sub-updates*, each consisting of a fixed value vector $v_i^l$ scaled by an input-dependent coefficient $m_i^l$. Crucially, the vectors $v_i^l$ are static model parameters, while all input dependence enters exclusively through the scalars $m_i^l$.

**Effect on Token Probabilities.** Let $E \in \mathbb{R}^{|\mathcal{V}| \times d}$ denote the unembedding matrix, and let $e_t \in \mathbb{R}^d$ be the row of $E$ corresponding to token $t \in \mathcal{V}$. Ignoring normalization constants, the probability of generating token $t$ after adding a single sub-update $m_i^l v_i^l$ to the residual stream can be written as

$$p(t \mid x^l + m_i^l v_i^l) \propto \exp\left(\langle e_t, x^l \rangle\right) \cdot \exp\left(\langle e_t, m_i^l v_i^l \rangle\right). \text{[2]} \quad (5)$$

Hence, the contribution of $v_i^l$ to the logit of token $t$ is additive and proportional to $\langle e_t, v_i^l \rangle$, scaled by $m_i^l$. If $\langle e_t, v_i^l \rangle > 0$, increasing $m_i^l$ raises the probability of token $t$, whereas $\langle e_t, v_i^l \rangle < 0$ suppresses it.

---

[2] Equation (5) characterizes the *local* contribution of a single MLP sub-update to token logits, abstracting away layer normalization, bias terms, and downstream residual interactions. As in prior mechanistic interpretability work (Elhage et al., 2021; Geva et al., 2022), we treat this decomposition as a first-order approximation that remains informative despite normalization and compositional effects across layers.

**Static Versus Input-Dependent Components.** The inner product $\langle e_t, v_i^l \rangle$ depends only on model parameters and is therefore independent of the input. All input-specific effects are mediated through the scalar activation $m_i^l = f(\langle k_i^l, x^l \rangle)$, which depends on the interaction between the residual stream representation $x^l$ and the corresponding key vector $k_i^l$. This separation allows us to interpret $v_i^l$ as encoding a *direction in representation space that promotes or suppresses specific tokens*, while $m_i^l$ determines how strongly this direction is activated for a given input.

### 4.2. Constructing Probes for Identifying Party MLP Value Vectors

As shown, MLP updates decompose into sums of input-dependent scaling coefficients applied to static value vectors. This decomposition implies a natural separation between (i) *which* directions in representation space promote party–related tokens and (ii) *how strongly* these directions are activated by a given input. Accordingly, our first objective is to identify value vectors whose directions in representation space are *predictively aligned* with tokens associated with a specific party.

**Probe Training on Intermediate Representations.** We focus on intermediate layers, which are known to encode high-level semantic and conceptual information more strongly than early or final layers that are optimized for next-token prediction (Panickssery et al., 2024). For each layer $l \in [\lfloor 0.5L \rfloor, \lceil 0.9L \rceil]$, we define $\bar{x}^l \in \mathbb{R}^d$ as the mean residual stream over all token positions in the sequence. We select this range based on a systematic ablation across relative layer windows (cf. Figure 7 in Appendix): the $0.5L - 0.9L$ window exhibits the highest lower whisker in the selectivity-gap distribution across seeds, models, and countries. We emphasize that this range should be treated as a tunable hyperparameter; our results identify it as a robust default rather than a universal optimum. We use mean pooling over token positions to obtain a sequence-level representation that is invariant to prompt length and surface phrasing. In preliminary analyses, alternative pooling strategies (e.g., first-token) yielded qualitatively similar probe directions.

For each party $o \in \mathcal{O}$, we train a linear probe to predict whether a residual representation corresponds to statements attributed to party $o$. Following prior mechanistic probing work (Lee et al., 2024), the probe computes a logit

$$z_o = W_n^\top \bar{x}^l, \qquad W_o \in \mathbb{R}^d, \quad (6)$$

where $\bar{x}^l$ denotes the mean-pooled residual stream at layer $l$. The predicted probability is given by the logistic sigmoid

$$\hat{y} = \sigma(z_o) = \frac{1}{1 + \exp(-z_o)}. \quad (7)$$

Probes are trained using a weighted binary cross-entropy loss with logits,

$$\mathcal{L} = -w_1 \, y \log \sigma(z_o) - (1-y)\log\bigl(1 - \sigma(z_o)\bigr), \quad (8)$$

with $y \in \{0, 1\}$ indicating whether the input is associated with party $o$ and $w_1$ correcting for class imbalance.

**Identifying Aligned and Diametric Value Vectors.** After training, the probe weight vector $W_o$ defines a direction in representation space that is predictive of party $o$. For each layer $l$ and MLP neuron $i$, we compute the cosine similarity

$$\cos(\theta_i^l) = \frac{\langle W_o, v_i^l \rangle}{\|W_o\| \, \|v_i^l\|}, \qquad i \in \{1, \ldots, d_{\mathrm{mlp}}\}. \quad (9)$$

Let $\mathcal{C}^l = \{\cos(\theta_i^l)\}_{i=1}^{d_{\mathrm{mlp}}}$ denote the distribution of cosine similarities in layer $l$. We compute the empirical first and third quartiles $Q_1^l$ and $Q_3^l$ and define the interquartile range $\mathrm{IQR}^l = Q_3^l - Q_1^l$. Value vectors are selected if their cosine similarity lies outside a 2.5-IQR fence. This criterion identifies value vectors whose alignment with the probe is statistically extreme relative to other MLP directions within the same layer, yielding a layer-adaptive, model-scale-invariant selection rule. We further distinguish between *probe-aligned* vectors

$$\hat{V}_+^n = \{v_i^l \in \hat{V}^n \mid \cos(\theta_i^l) > 0\} \quad (10)$$

and *diametric* vectors

$$\hat{V}_-^o = \{v_i^l \in \hat{V}^o \mid \cos(\theta_i^l) < 0\}, \quad (11)$$

which respectively promote or suppress party-related evidence, thereby capturing both supportive and diametrical contributions to the latent representation of the probe concept.

To ensure that selected value vectors contribute to party token generation, we perform a sign-inversion–based validation. For each $v_i^l \in \hat{V}^o$, we counterfactually flip the corresponding sub-update in the residual stream by reversing its sign and measure the resulting change in the log-probability of a party token $t_o$:

$$\Delta \log p(t_o) = \log p(t_o \mid x^l) - \log p(t_o \mid x^l - 2m_i^l v_i^l). \quad (12)$$

This sign-inversion criterion provides a *local necessity test*: a value vector is retained only if its removal decreases the median log-probability of the corresponding party token on a held-out test dataset (Figure 8 in Appendix). The resulting sets $\hat{V}_+^o$ and $\hat{V}_-^o$ represent, respectively, static value vectors that promote and suppress party-related tokens. To complement this selection criterion with a functionally stronger check, we additionally evaluate the selected vectors in a controlled non-political intervention setup (cf. Figure 9 in Appendix): on a dataset of 1,280 non-political persona–question pairs (e.g., "What ice cream flavor would I order?"),

scaling the selected value vectors significantly increases the party-token logit relative to a no-scaling baseline (paired $t$-test, $p < 0.001$), whereas a matched random-vector control of equal magnitude shows no significant effect. Consistent with this, preliminary experiments without the sign-inversion gating did not produce consistent intervention effects.

### 4.3. Persona-Induced Activation of Party Value Vectors

To analyze how persona descriptions are mapped to party representations, we construct a controlled set of persona prompts and measure how they activate the party–related value vectors identified in the previous subsection.

**Persona Construction and Prompt Variation.** As described in Section 3, each persona $p \in \mathcal{P}$ is defined as a combination of socio-demographic and ideological attributes (cf. Table 1 in Appendix). To account for prompt sensitivity, we instantiate each persona using $J = 10$ independently designed prompt templates, yielding a set of prompts $\{(p, j) : p \in \mathcal{P}, j \in \mathcal{J}\}$, comprising $280,000$ persona combinations. Attribute combinations are sampled such that the empirical distribution of personas matches the marginal distributions observed in the corresponding real-world survey, ensuring comparability between synthetic and human data.

**Measuring Value-Vector Activations.** For each prompt $(p, j)$, we run inference and record, for all layers $l$ and all party–related value vectors $v_i^l \in \hat{V}^o$, their input-dependent scaling coefficients $m_{i,p,j}^l$. To account for heterogeneous alignment strengths, we weight these activations by their cosine similarity with the party probe:

$$a_{i,p,j}^l = m_{i,p,j}^l \cdot \cos(\theta_i^l), \quad (13)$$

where $\cos(\theta_i^l)$ is defined in Equation (9). Specifically, activations are normalized within each layer across all personas and prompt variants.

**Aggregating Party Activation Scores.** For each party $o$, persona $p$, and prompt variant $j$, we define an aggregated activation score as

$$A_{p,j}^o = \frac{1}{|\hat{V}^o|} \sum_{v_i^l \in \hat{V}^o} a_{i,p,j}^l, \quad (14)$$

which captures the extent to which persona $p$ activates party–related directions in the model's latent space. This yields a multivariate activation vector $A_{p,j} = (A_{p,j}^{o_1}, \ldots, A_{p,j}^{o_{|\mathcal{O}|}})$ for each persona–prompt pair.

**Group-Level Aggregation.** To study systematic patterns, we aggregate activation vectors across persona attributes. Let $k$ index a persona attribute (e.g., age, employment), and let $\mathcal{G}_k$ denote the set of its categorical values. For

each category $g \in \mathcal{G}_k$, let $\mathcal{P}_{k,g} \subseteq \mathcal{P}$ denote the subset of personas whose attribute $k$ takes value $g$. We define the latent activation–based distribution over categories of attribute $k$ as

$$\Psi_k = \left(\Psi_{k,g}\right)_{g \in \mathcal{G}_k}, \quad \Psi_{k,g} = \mathbb{E}_{p \sim \mathcal{P}_{k,g}, \, j \sim \mathcal{J}} \left[A_{p,j}\right], \quad (15)$$

where the expectation is taken with sampling weights that mirror the empirical category frequencies observed in the real-world survey. The resulting vector $\Psi_k$ defines a normalized distribution over the categories of attribute $k$ and can be interpreted as a Monte Carlo estimator of the expected latent party activation signal induced by that attribute. These attribute-level distributions form the basis for our comparison between latent activation-based representations and real-world surveys.

## 4.4. Distributional Comparison Between LLMs and Survey Data

To compare latent activation-based distributions with real-world surveys, we operate on attribute-level party distributions $\Psi_k$ defined in Equation (15). For each attribute $k$ and party $o$, we construct three normalized distributions: (i) $\Psi_k^{\text{latent}}$, derived from value-vector activations, (ii) $\Psi_k^{\text{prob}}$, derived from next-token probabilities, and (iii) $\Psi_k^{\text{survey}}$, derived from weighted survey responses.

**Choice of Distance Metrics.** We compare these distributions using two complementary metrics, selected according to the structure of the persona attribute. For *nominal attributes* (e.g., gender or education), we use the Jensen–Shannon (JS) distance $D_{\text{JS}}(P, Q)$, which is symmetric, bounded, and well-defined for empirical distributions. For *ordinal attributes* with a natural ordering (e.g., age or income), we use the first Wasserstein distance $D_{\text{W}}(P, Q)$. Unlike JS distance, Wasserstein distance accounts for the magnitude of shifts along the attribute axis, penalizing mass transport proportionally to its distance.

**Evaluation Protocol.** For each persona attribute $k$, party $o$, model, and country, we compare attribute-level preference distributions derived from LLMs and survey data. Specifically, we compute

$$D_k^{\text{latent}} = D\left(\Psi_k^{\text{latent}}, \Psi_k^{\text{survey}}\right), \quad D_k^{\text{prob}} = D\left(\Psi_k^{\text{prob}}, \Psi_k^{\text{survey}}\right), \quad (16)$$

where $D(\cdot, \cdot)$ denotes the distance metric appropriate for the attribute type. We define the distance difference as

$$\Delta_k = D_k^{\text{prob}} - D_k^{\text{latent}}. \quad (17)$$

A latent activation-based estimation is said to achieve an attribute-level win if $\Delta_k > 0$. Win-rates are computed as the proportion of attributes $k$ for which this condition holds, evaluated separately by model, country, and party (cf. Figure 11 for the distribution of $\Delta_k$ in the Appendix).

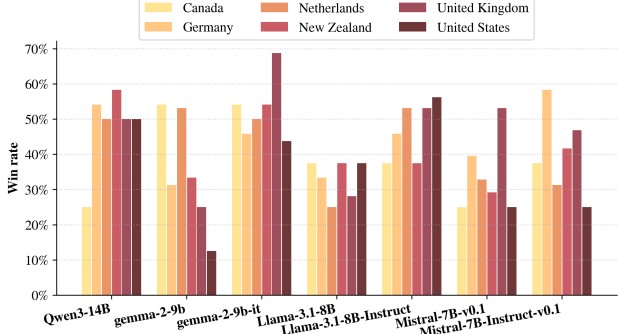

*Figure 2.* Win-rates by model and country comparing mechanistic forecasting ($\Psi_g^{\text{latent}}$) and probability-based ($\Psi_g^{\text{prob}}$) preference distributions against survey data ($\Psi_g^{\text{survey}}$).

## 5. Results

In the following, we first discuss the characteristics of the political party probes that we trained (Section 5.1). We then compare the performance of mechanistic forecasting against probability-based estimation across models and countries (Section 5.2). While this comparison demonstrates that mechanistic forecasting can improve voting-outcome predictions in many countries, we analyze political party-level (Section 5.3) and attribute-level differences (Section 5.4) to understand the specific patterns driving cross-national variation. These analyses help practitioners understand when mechanistic forecasting serves as a beneficial complement to probability-based estimation.

### 5.1. Probes Capture Political Party Associations

A prerequisite for mechanistic forecasting is that probes reliably identify party-associated structure in the model's internal representations. Across all our value probes, we achieve strong generalization performance on held-out test data: Probe F1 scores consistently exceed $96\%$ on a $10\%$ hold-out split. To verify that these probes capture genuine party-associated structure rather than arbitrary linear projections, we evaluate them against three control conditions and two regression baselines (cf. Figure 6 in Appendix). Probes with randomly shuffled labels (macro-F1 $\approx 0.33$) and random-weight probes (macro-F1 $\approx 0.24$) perform near chance, confirming that probe effectiveness depends on meaningful alignment between representations and party labels. Regression baselines trained on the same activations achieve strong but consistently lower performance ($0.93 \pm 0.05$) than our value probe ($0.99 \pm 0.01$; $p < 0.001$), indicating that the probe learns a more discriminative, task-aligned direction than a generic linear readout. Party associations are illustrated by projecting the identified value vectors into vocabulary space and inspecting the highest–cosine-similarity tokens (cf. Table 4, Appendix).

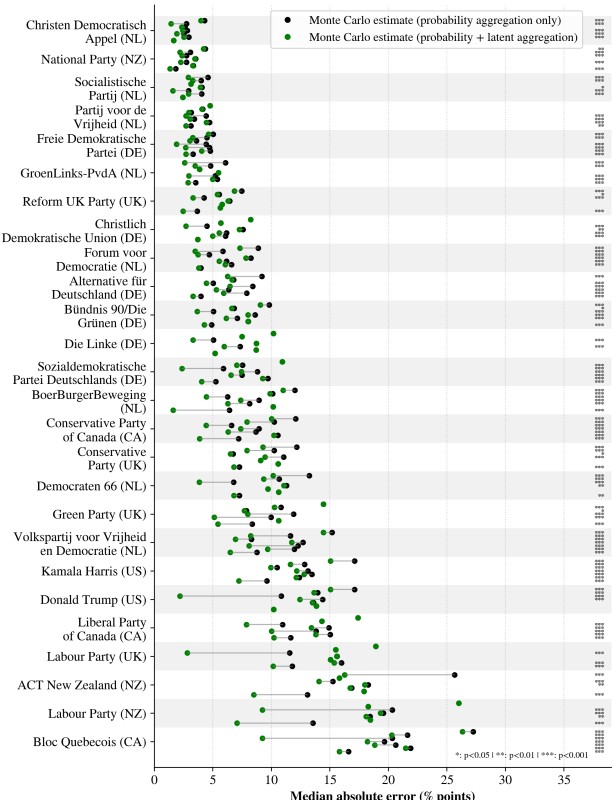

*Figure 3.* Party-level estimation error for estimating conditional vote shares. Each point shows the median absolute error in estimating $P(\text{party} \mid \text{category})$ relative to survey benchmarks, with offsets indicating different models. Green points highlight the potential gains achievable by choosing mechanistic forecasting estimates when they outperform probability-based estimations.

## 5.2. Mechanistic Forecasting Improves Predictions Across Models and Countries

Figure 2 provides an overview of win-rates for each model by country (cf. Figure 11 in Appendix for the full distribution of distance differences $\Delta_k$ underlying these win-rates). While performance varies across models and national contexts, the results consistently show that estimations of persona preferences derived from mechanistic forecasting distributions ($\Psi_g^{latent}$) can be leveraged to more closely predict real-world survey outcomes ($\Psi_g^{survey}$) than estimations based on final output token probabilities ($\Psi_g^{prob}$). If we compare across fine-tuned to base model versions, we observe that the win-rates increase even further for most countries and across models, indicating that the alignment process shifts estimations away from real-world survey predictions, making our latent approach more efficient. We further verify that these gains are not explained by access to the external party-position data alone: an in-context learning baseline that receives the same data through the prompt often remains worse than our mechanistic forecasting across models (cf.

Figure 10 in Appendix). To examine the sources of these cross-country differences, we analyze party-level estimation errors next.

## 5.3. Political Party Differences

Figure 3 reports party-level estimation error for estimating party vote shares conditional on persona categories. Specifically, we evaluate the absolute error in predicting $P(\text{party} \mid \text{category})$, comparing LLM-based estimates to survey benchmarks. These errors correspond to party-wise marginal projections of the attribute-level distributions $\Psi_k$ used in our distributional evaluation. Each point corresponds to an LLM-based estimation aggregated over Monte Carlo samples, with separate offsets indicating different models and thus distinct induced probability distributions. Across parties and models, estimation error varies substantially, reflecting heterogeneity in how well party-specific vote shares can be recovered from persona information. The dispersion of points illustrates that mechanistic forecasting yields systematically different outcomes, even when targeting the same party–category relationship. Importantly, for a subset of categories, mechanistic forecasting estimates—derived from Monte Carlo aggregation over party-aligned latent activations induced by sampled personas—produce predictions that are closer to survey outcomes than those obtained from next-token probability–based estimates alone. These potential improvements are highlighted by the green points, which indicate reduced absolute error relative to probability-based baselines. Three patterns emerge: First, for nearly all investigated parties and models, there exist mechanistic forecasting estimators that improve party-share predictions for at least some categories. Second, overall error levels are broadly comparable across models, suggesting similar aggregate performance despite architectural and training differences. Third, the contribution of latent information becomes increasingly variable as estimation error grows, indicating that latent signals matter most for parties that are harder to predict from probabilities alone. That these signals are functional is supported by our intervention analysis (cf. Figure 9 in Appendix)

## 5.4. Persona Attribute Differences

Figure 4 compares mechanistic forecasting estimators and probability-based estimators across persona attributes and countries, aggregated over models. Two systematic patterns emerge: First, latent aggregation yields the largest and most consistent gains for *demographic attributes*, particularly in the United States and the United Kingdom. Age, education, and employment exhibit high win-rates in both countries, indicating that demographic information is more reliably recovered from latent representations than from surface-level output probabilities. Household income shows more heterogeneous behavior, with strong improvements

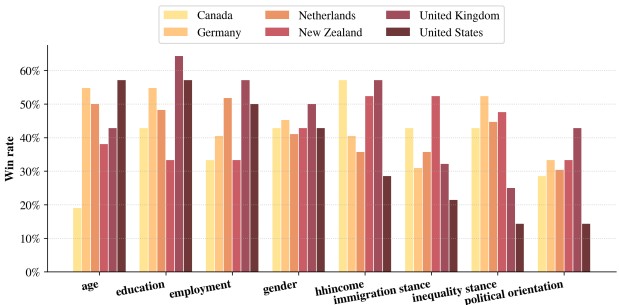

*Figure 4.* Latent win-rates by persona attribute and country, aggregated across models. Each bar reports the fraction of cases in which mechanistic forecasting is closer to survey estimates of category shares conditional on party than probability-based estimations.

in Canada and mixed performance in the United States. In contrast, demographic attributes are less well captured in New Zealand, while Germany and the Netherlands exhibit comparatively uniform performance across demographic categories. Second, *opinion-based attributes* display greater cross-national variation. Latent gains for political orientation are strongest in the United Kingdom but substantially weaker in the Netherlands. For issue positions, Germany shows comparatively low win-rates for immigration despite strong demographic performance, whereas New Zealand exhibits the opposite pattern, with higher gains for immigration and inequality stance. These differences suggest that the usefulness of mechanistic forecasting for opinion-based attributes depends strongly on how explicitly such dimensions are expressed in surface-level predictions within each national context. Overall, mechanistic forecasting improves attribute-level preference estimation in many settings, but its benefits are selective rather than uniform. Attributes that encode politically relevant information in a diffuse manner (notably demographics) benefit most from latent representations, while attributes that are already salient at the output level exhibit more variable gains across countries (cf. Table 1 in Appendix).

### 5.5. Entropy and Predictive Performance

A central question raised by our results is *when* mechanistic forecasting should be preferred over probability-based estimations. While the existence of exploitable latent structure is informative in itself, its practical relevance depends on identifying settings in which latent estimators provide systematic advantages. Our analyses suggest that attribute-level heterogeneity provides a useful criterion. Specifically, we find that mechanistic forecasting estimators are most effective for persona attributes with high normalized entropy, where no single category dominates, and accurate prediction requires aggregating weak but distributed signals. This effect is particularly pronounced for instruction-tuned

models, for which surface-level output probabilities tend to concentrate mass on a small subset of categories. To formalize this intuition, we estimate a regression model in which the outcome is the attribute-level distance difference $\Delta_k$, focusing on cases with $\Delta_k > 0$. We relate $\Delta_k$ to the normalized entropy of attribute-level probability distributions induced by LLM outputs and selectively evaluate mechanistic forecasting estimations only for attributes with normalized entropy exceeding $0.85$. Figure 5 shows the resulting median improvement in prediction error, reported separately by attribute and model. Across attributes, entropy-gated filtering reveals consistent gains from mechanistic forecasting estimation, with improvements emerging most clearly in high-entropy regimes where probability-based estimations are least informative. Importantly, these results do not imply that latent estimators universally dominate probability-based approaches. Rather, they indicate that latent representations provide more reliable estimates of conditional distributions—such as $P(\text{category} \mid \text{party})$—precisely when surface-level probabilities are diffuse and uncertain. In low-entropy settings, where probability-based estimations already concentrate mass on a small number of categories, gains from latents are correspondingly limited.

## 6. Discussion and Conclusion

Our results suggest that a limiting part of existing LLM-based preference prediction methods lies not in the *absence* of relevant information, but in how this information is *elicited*. Across models, countries, and persona attributes, we find that latent representations encode systematic signals about political persona preferences that are often distorted or suppressed in surface-level output probabilities. By aggregating political party-aligned latent activations, our suggested method of mechanistic forecasting extends preference prediction from a prompting problem to a representation-aware estimation problem. This shift has implications beyond the electoral setting studied here, highlighting the gain of leveraging internal model structure when LLMs are used to estimate collective human preferences.

**When and Why Latents can Help.** The gains from mechanistic forecasting are not uniform. We observe the greatest improvements for demographic attributes and in high-entropy settings where output probabilities are diffuse and unstable. In contrast, for low-entropy attributes, mechanistic forecasting offers limited additional benefit. These patterns suggest that latent signals function as weak but distributed indicators that become informative when surface probabilities collapse or overconcentrate, particularly in instruction-tuned models. From a practical perspective, this implies that mechanistic forecasting should be applied selectively and guided by diagnostic indicators such as attribute-level entropy.

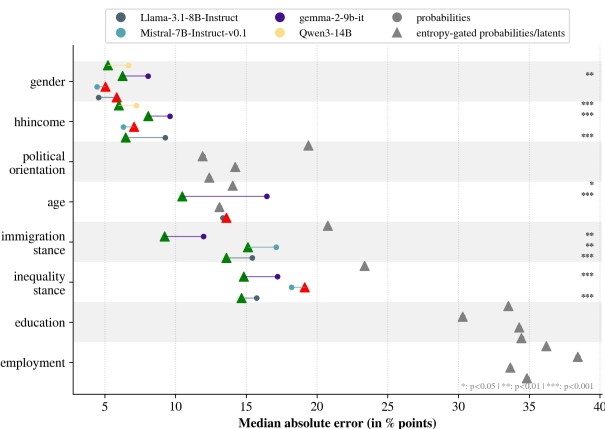

*Figure 5.* Median probability-based estimation error and corresponding improvement from substituting mechanistic forecasting estimates for category–given–party probabilities, evaluated on high-entropy attributes ($> 0.85$). Green indicates error reduction, red indicates increased error, and gray indicates no change.

**Interpreting Latent Preference Signals.** Importantly, our findings do not imply that LLMs "hold" preferences or beliefs. Latent activations reflect learned statistical associations between persona attributes and political outcomes encoded during training, rather than normative or causal judgments. The selected value vectors nonetheless exert a direct functional influence on party-token generation (cf. Figure 9 in Appendix), and the gains we observe are not explained by access to the external party-position data alone (cf. Figure 10 in Appendix). Compared to output probabilities, latent activations retain weak but distributed internal associations that become informative under Monte Carlo aggregation, particularly in high-entropy settings where surface-level probability estimates are more diffuse. We therefore view latent and surface-level signals as complementary: output probabilities often capture sharp, alignment-driven predictions, while latent activations can preserve distributed internal evidence that would otherwise be lost at the decoding stage.

**Implications for Social Science Applications.** From a social science perspective, mechanistic forecasting should be understood as a complementary tool rather than a substitute for surveys. Traditional surveys remain essential for capturing preferences of underrepresented populations or in high-stakes decisions requiring precise measurement. However, when used responsibly, mechanistic forecasting offers a novel way to extract population-level signals if practitioners turn to LLMs for predicting human preferences.

**Limitations and Broader Implications.** Our approach requires white-box access to model internals and computational resources to train probes, limiting its applicability in some settings. In addition, our analysis relies on a local, first-order approximation of MLP contributions to token logits, which may miss higher-order interactions introduced by

normalization and residual composition across layers. Another methodological choice concerns our use of per-party binary probes; training a multiclass linear head and using each class-specific weight vector in place of $W_o$ is a natural extension that optimizes joint discrimination across parties, which we leave to future work. Despite these limitations, the methodological framework we develop is applicable beyond elections: any domain with structured, categorical preferences—including consumer choice, values or policy attitudes—can in principle benefit from similar latent-based aggregation. More broadly, our results position social science prediction as a challenging testbed for interpretability methods, extending prior work on hidden knowledge from binary factual correctness to complex, distributional outcomes. Understanding when and how internal representations support reliable aggregation remains an important direction for future research.

## Impact Statement

This work studies whether LLM internal representations encode information about human political preferences, finding that latent activations can carry exploitable signal not fully reflected in model outputs, which suggests that LLMs may encode more about human preferences than their surface predictions reveal. Used responsibly, this can support exploratory social-science analysis. However, methods that extract such signals from model internals also risk misuse, such as population profiling or micro-targeted political messaging, and may reflect demographic and cross-national biases in the training data. We therefore stress that our findings do not imply LLMs hold political beliefs, and that mechanistic forecasting is a complement to, not a substitute for, representative surveys.

## Acknowledgements

Part of this work was supported by the DAAD programme Konrad Zuse Schools of Excellence in Artificial Intelligence, sponsored by the German Federal Ministry of Education and Research (SB). The authors also gratefully acknowledge the scientific support and HPC resources provided by the Erlangen National High Performance Computing Center (NHR@FAU) of the Friedrich-Alexander-Universität Erlangen-Nürnberg (FAU). The hardware is funded by the German Research Foundation (DFG). This work was done in part while SB and FK were visiting the Simons Institute for the Theory of Computing.

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

# A. Further Information on Constructed Personas and Attributes

This section documents the persona attribute schema used to instantiate synthetic survey respondents and the prompt templates that embed these attributes into model inputs.

**Persona Attributes.** Table 1 lists the persona attributes, their categorical values per language, and their measurement scale (ordinal or nominal). Attribute categories follow established voting predictors from political science and are drawn from representative election surveys per country.

*Table 1.* Persona attributes and characteristics

| ATTRIBUTE | ENGLISH | GERMAN | DUTCH | SCALE |
|---|---|---|---|---|
| Age | younger than 20; 20–29; 30–39; 40–49; 50–59; 60–69; 70+ | jünger als 20; 20–29; 30–39; 40–49; 50–59; 60–69; älter als 70 | jonger dan 20; 20–29; 30–39; 40–49; 50–59; 60–69; ouder dan 70 | ordinal |
| Gender | male; female | männlich; weiblich | mannelijk; vrouwelijk | nominal |
| Education | no qualification; high school; college; university degree | kein Abschluss; Hauptschule; Realschule; Abitur; Hochschulabschluss | basisonderwijs; beroepsopleiding; havo/vwo; hbo/universiteit | ordinal |
| Household income | low; middle; high | niedrig; mittel; hoch | laag; gemiddeld; hoog | ordinal |
| Employment | working; in training; student; retired; not working | berufstätig; in Ausbildung; nicht berufstätig | werkzaam; in opleiding; gepensioneerd; niet werkzaam | nominal |
| Political ideology | strongly right; center-right; center; center-left; strongly left | stark rechts; rechts der Mitte; Mitte; links der Mitte; stark links | sterk rechts; rechts van het midden; midden; links van het midden; sterk links | ordinal |
| Immigration stance | fewer; same; more; agree; disagree | einschränken; weder noch; erleichtern | eens; niet eens | ordinal |
| Inequality stance | disagree; neutral; agree | dagegen; unentschlossen; dafür | tegen; onbeslist; voor | ordinal |

**Persona Prompt Templates.** Table 2 shows three of the ten persona prompt variants used per country, illustrating the wording and sentence-order variation introduced to account for prompt sensitivity. As described in Section 3, all ten variants were hand-crafted in German and translated into the other languages with native-speaker verification; placeholders in braces (e.g., {age}, {political_orientation}) are substituted with persona attribute values at runtime. Note that the translations also take into account wording differences between the respective country surveys. The complete set of variants per country is released in our code repository[1].

*Table 2.* Exemplary prompt schemes of different countries

| INDEX | UNITED STATES | NETHERLANDS | GERMANY |
|---|---|---|---|
| 1 | *I am {age} years old and {gender}. I have {education}, my household income is {hhincome}, and I am {employment}. Ideologically, I lean towards {political_orientation}. On immigration and inequality, my views are {immigration} and {inequality}. If elections were held in {year_of_election}, which party would I vote for? I vote for the party ...* | *Ik ben age jaar en {gender}. Ik heb {education} gevolgd, mijn netto maandinkomen is {hhincome} en ik ben {employment}. Ideologisch neig ik naar de positie {political_orientation}. Ik ben het {immigration} met de stelling dat mijn gemeente opvang moet bieden aan asielzoekers als dat nodig is en ik ben {inequality} ten aanzien van overheidsmaatregelen om inkomensverschillen te verkleinen. Als er {year_of_election} verkiezingen zouden zijn, op welke partij zou ik dan stemmen? Ik stem op de partij...* | *Ich bin {age} Jahre alt und {gender}. Ich habe {education}, mein monatliches Nettoeinkommen im Haushalt ist {hhincome} und ich bin {employment}. Ideologisch neige ich zur Position {political_orientation}. Ich denke, die Regierung sollte den Zuzug von Ausländern {immigration} und bin {inequality} in Bezug auf staatliche Maßnahmen zur Verringerung von Einkommensunterschieden. Wenn {year_of_election} Wahlen wären, für welche Partei würde ich stimmen? Ich wähle die Partei...* |
| 2 | *I believe the number of immigrants from foreign countries who are permitted to come to the United States to live should be {immigration} and I {inequality} that the government should see to jobs and standard of living. Ideologically, I lean toward the {political_orientation} position. I have {education}, my household's monthly net income is {hhincome}, and I am {employment}. I am {age} years old and {gender}. If there were elections {year_of_election}, which party would I vote for? I vote for the party...* | *Ik ben het {immigration} met de stelling dat mijn gemeente opvang moet bieden aan asielzoekers als dat nodig is en ik ben {inequality} ten aanzien van overheidsmaatregelen om inkomensverschillen te verkleinen. Ideologisch neig ik naar de positie {political_orientation}. Ik heb {education} gevolgd, mijn netto maandinkomen is {hhincome} en ik ben {employment}. Ik ben {age} jaar en {gender}. Als er {year_of_election} verkiezingen zouden zijn, op welke partij zou ik dan stemmen? Ik stem op de partij...* | *Ich denke die Regierung sollte den Zuzug von Ausländern {immigration} und bin {inequality} in Bezug auf staatliche Maßnahmen zur Verringerung von Einkommensunterschieden. Ideologisch neige ich zur Position {political_orientation}. Ich habe {education}, mein monatliches Nettoeinkommen im Haushalt ist {hhincome} und ich bin {employment}. Ich bin {age} Jahre alt und {gender}. Wenn {year_of_election} Wahlen wären, für welche Partei würde ich stimmen? Ich wähle die Partei...* |
| 3 | *In terms of age, I am {age} and my gender is {gender}. I have {education}, my monthly household net income is {hhincome}, and I am {employment}. Politically, I lean toward the {political_orientation} position. When asked about immigration, I say the number of immigrants permitted to the US should be {immigration}. Also, I {inequality} when it comes to whether the government should see to jobs and standard of living. Which party would I choose in an election in {year_of_election}? I choose the party...* | *Wat betreft mijn leeftijd ben ik {age} jaar en mijn geslacht is {gender}. {education} heb ik gevolgd, mijn netto maandinkomen is {hhincome}, en ik ben {employment}. Politiek gezien neig ik naar de positie {political_orientation}. Als men mij vraagt naar mijn mening over immigratie, ben ik het [eens/niet eens] met de uitspraak dat mijn gemeente indien nodig opvang moet regelen voor asielzoekers. Daarnaast ben ik {inequality} over de vraag of de overheid maatregelen moet nemen om inkomensverschillen te verkleinen. Voor welke partij zou ik kiezen bij een verkiezing in {year_of_election}? Ik kies de partij...* | *Bezogen auf mein Alter bin ich {age} und mein Geschlecht ist {gender}. {education} habe ich, mein monatliches Haushaltsnettoeinkommen ist {hhincome}, und ich bin {employment}. Politisch gesehen neige ich zur Position {political_orientation}. Fragt man mich zu meiner Meinung bezüglich Immigration, sage ich, dass man sie {immigration} soll. Außerdem bin ich {inequality} was die Frage angeht, ob die Regierung Maßnahmen ergreifen sollte, um die Einkommensunterschieden zu verringern. Für welche Partei würde ich mich bei einer Wahl im Jahr {year_of_election} entscheiden? Ich wähle die Partei...* |

## B. Further Information on Value Probes and Value Vectors

### B.1. Value-Probe Prompt Templates

Table 3 shows three of the ten value-probe prompt variants used to extract party-aligned activations from voting-advice material (statement, party answer, party comment) during probe training, in English, German, and Dutch. As with the persona prompts (cf. Table 2), the German variants were hand-crafted and translated into the remaining languages with native-speaker verification; the placeholders {statement}, {answer}, {comment}, and {shuffled-party-list} are substituted with the corresponding voting-advice item and the shuffled candidate party-name list at runtime. The complete set of ten variants per country is released in our code repository[1].

*Table 3.* Exemplary value-probe prompt variants used during probe training, shown in English, German, and Dutch. The full set of ten variants per country is provided in our code repository.

| INDEX | ENGLISH | GERMAN | DUTCH |
|---|---|---|---|
| 1 | *You are given a political statement, a response to it, and a comment. These are from a political party in the United Kingdom.* **Statement:** *{statement}* **Response:** *{answer}* **Comment:** *{comment}* *Choose from the following list: {shuffled-party-list}. Which party gave this response and comment? Respond only with the name of the party.* | *Du bekommst eine politische Aussage, eine Antwort darauf und einen Kommentar. Diese stammen von einer deutschen Partei.* **Aussage:** *{statement}* **Antwort:** *{answer}* **Kommentar:** *{comment}* *Wähle aus der folgenden Liste: {shuffled-party-list}. Welche Partei hat diese Antwort und diesen Kommentar abgegeben? Bitte antworte nur mit dem Parteinamen.* | *Je krijgt een politieke uitspraak, een reactie daarop en een toelichting. Deze zijn afkomstig van een Nederlandse partij.* **Uitspraak:** *{statement}* **Antwoord:** *{answer}* **Toelichting:** *{comment}* *Kies uit de volgende lijst: {shuffled-party-list}. Welke partij heeft deze reactie en toelichting gegeven? Antwoord alleen met de naam van de partij.* |
| 2 | *Read the following political statement and the response from an anonymous party. Your task is to identify which party made the response and comment.* *### Statement: {statement}* *### Response: {answer}* *### Comment: {comment}* *Which party from the list {shuffled-party-list} fits best? Provide only the party name as your answer.* | *Lies die folgende politische Aussage und die Reaktion einer anonymen Partei darauf. Deine Aufgabe ist es, zu erkennen, welche Partei hinter der Antwort und dem Kommentar steckt.* *### Aussage: {statement}* *### Antwort: {answer}* *### Kommentar: {comment}* *Welche Partei aus der Liste {shuffled-party-list} trifft am ehesten zu? Gib nur den Parteinamen als Antwort an.* | *Lees de volgende politieke uitspraak en de reactie van een anonieme partij daarop. Jouw taak is te bepalen welke partij achter het antwoord en de toelichting zit.* *### Uitspraak: {statement}* *### Antwoord: {answer}* *### Toelichting: {comment}* *Welke partij uit de lijst {shuffled-party-list} past hier het best bij? Geef enkel de naam van de partij als antwoord.* |
| 3 | *Imagine you're playing a political matching game. You're given a comment, a response, and the related political statement.* *{statement}* *{answer}* *{comment}* *Choose the party that most likely expressed this from the list: {shuffled-party-list}. Provide only the party name.* | *Stelle dir vor, du wärst in einem politischen Zuordnungsspiel. Du bekommst einen Kommentar, eine Antwort und die dazugehörige politische Aussage.* *{statement}* *{answer}* *{comment}* *Wähle die Partei, die das geäußert haben könnte, aus dieser Liste: {shuffled-party-list}. Nenne nur den Parteinamen.* | *Stel je voor dat je een politieke matching-quiz speelt. Je krijgt een toelichting, een antwoord en de bijbehorende uitspraak.* *{statement}* *{answer}* *{comment}* *Kies de partij die dit het best verwoord zou kunnen hebben, uit deze lijst: {shuffled-party-list}. Noem enkel de partijnaam.* |

## B.2. Value-Probe Validation

We validate the value probes along two dimensions: (i) that the extracted signal is not recoverable from *trivial* baselines, and (ii) that the chosen layer range is empirically justified rather than arbitrary.

**Comparison Against Control and Regression Baselines.** Figure 6 compares our value probe against two control conditions and two regression baselines trained on the same activations. The *label-shuffled control* trains a probe on randomly permuted party labels, destroying any genuine association between representations and parties; the *random-weight control* uses untrained, randomly initialized probe directions, testing whether arbitrary directions achieve comparable performance. The regression baselines replace classification with regression on the same features, using either binary hard labels or continuous soft targets. Our probe outperforms all four: the label-shuffled and random-weight controls perform near chance, while the regression baselines remain below our probe. This indicates that the extracted signal is not recoverable from shuffled supervision, random directions, or generic linear readout, supporting that our value probe captures meaningful party-associated structure in model representations.

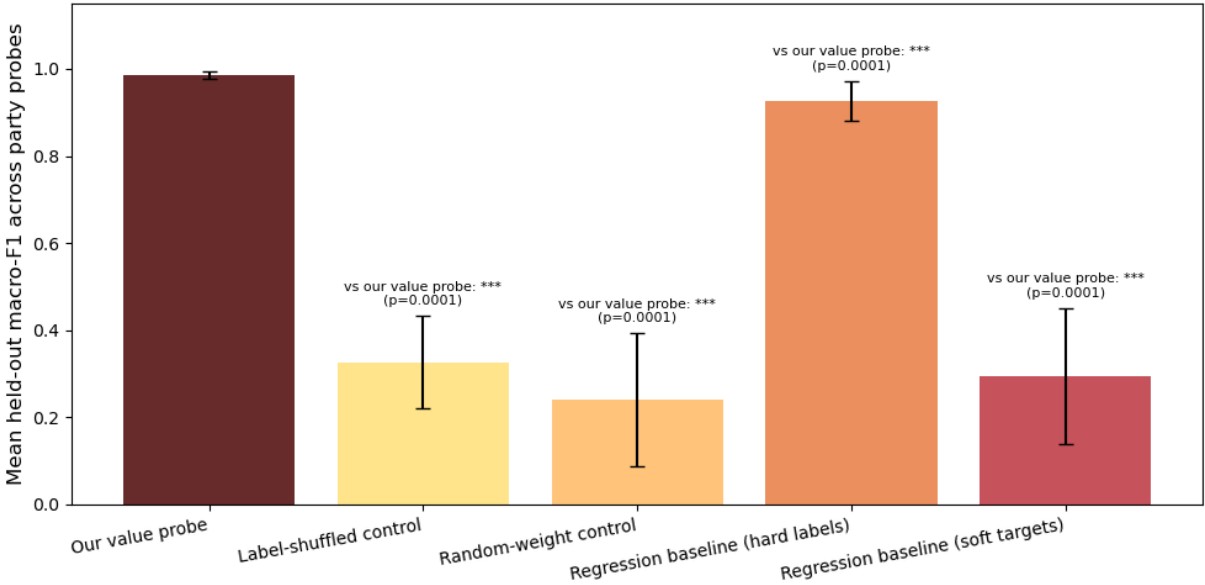

*Figure 6.* Held-out performance of party probes under control and regression baselines. Bars show mean held-out macro-F1 across one-vs-rest party probes for our value probe, compared to three control conditions and two regression baselines. The *label-shuffled control* is trained on randomly permuted labels, removing any true association between representations and parties. The *random-weight control* uses randomly initialized probe directions without training, testing whether arbitrary directions can achieve similar performance. The regression baselines replace classification with regression on the same features: (i) *regression on binary hard labels* (party vs. rest) and (ii) *regression on continuous soft targets* reflecting probabilistic party support. Results are averaged across all countries and models, and across multiple probe configurations, including random seeds (0–9), layer ranges (0.4–0.7 and 0.8–1.0 of model depth), a 10% held-out test split, and 4000 training epochs. Error bars denote standard deviation across configurations. Significance annotations report permutation tests comparing each baseline to our value probe. The performance gap between our value probe and all baselines indicates that the extracted signal is less recoverable from shuffled supervision, random directions, or regression-only mappings, supporting that our value probe captures meaningful party-associated structure in model representations.

**Layer-Window Selection.** Figure 7 reports the selectivity gap $\Delta_{\mathrm{sel}}$ across relative layer windows, motivating our default choice of $0.5L$–$0.9L$ in Section 4.2. We treat the layer range as a tunable hyperparameter; the analysis identifies $0.5L$–$0.9L$ as a stable default rather than a universal optimum.

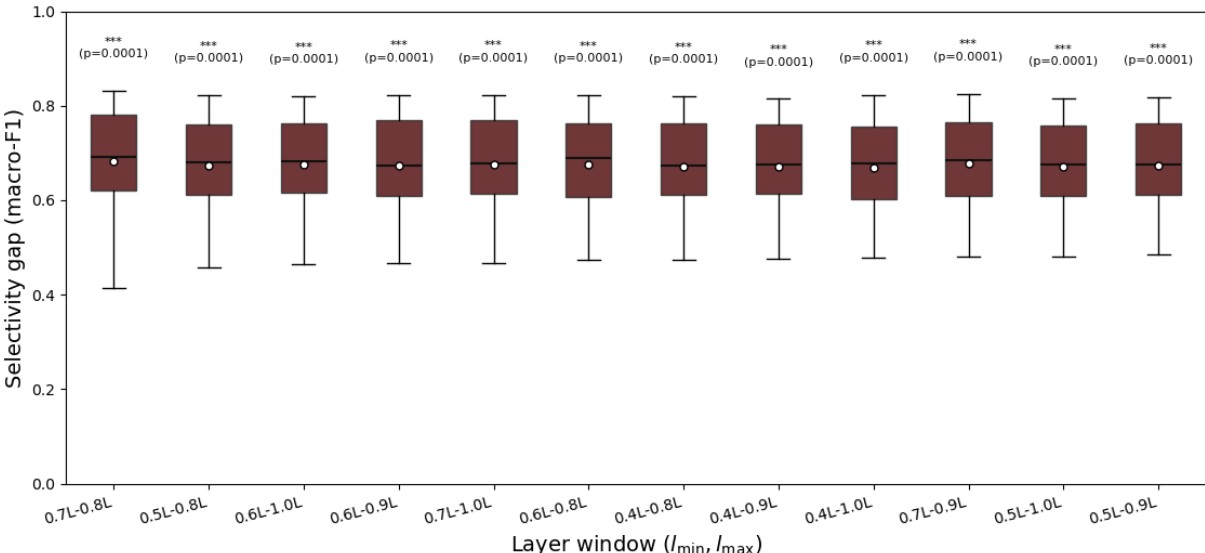

*Figure 7.* Selectivity gap across relative layer windows, aggregated over all countries and models. Each boxplot shows the distribution of seed-level selectivity gaps $\Delta_{\mathrm{sel}} = \mathrm{MacroF1}_{\mathrm{real}} - \mathrm{MacroF1}_{\mathrm{shuffle}}$ for a given relative layer window $(l_{\min}, l_{\max})$, where macro-F1 is first averaged over parties within each seed (0–9) and then aggregated across seeds. The x-axis reports the tested layer windows as fractions of total model depth $L$. Boxes indicate the interquartile range (IQR), center lines denote medians, whiskers follow the standard $1.5\times$ IQR rule, and white markers indicate means. Significance annotations report one-sided sign-flip tests against zero. Among the evaluated windows, the range $0.5L$–$0.9L$ exhibits the highest lower whisker of the selectivity-gap distribution, indicating the most consistently positive separation between our value probe and the label-shuffled control across seeds. We therefore use this intermediate-to-late layer range as the default probe-search region in the main experiments.

## B.3. Value-Vector Selection

Figure 8 illustrates the value-vector selection procedure described in Section 4.2: cosine similarities between the trained party probe and MLP value vectors are computed per layer, IQR-extreme directions are identified as candidate probe-aligned and diametric vectors, and only those whose sign inversion decreases the median log-probability of the corresponding party token are retained.

## B.4. Top Tokens of Value Probes and Value Vectors

Table 4 projects the trained value probes and the selected value vectors into vocabulary space and lists the highest-cosine-similarity tokens per party for Qwen3-14B. Some directions surface semantically interpretable tokens (e.g., "left" tokens for *Die Linke* and *Kamala Harris*, "socialist"-related tokens for the *New Zealand Labour Party*), while others remain less directly interpretable, consistent with the distributed nature of latent party-relevant signals.

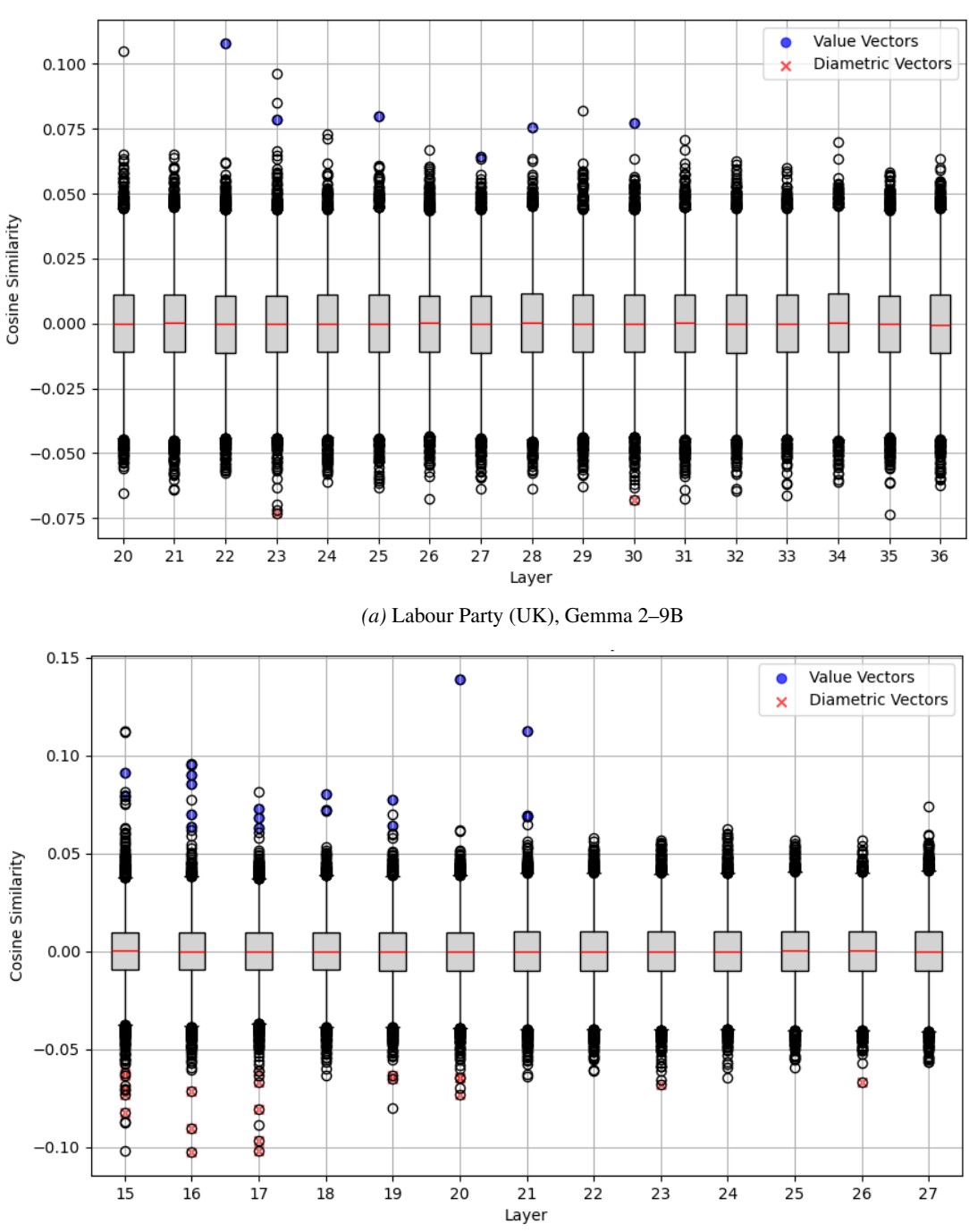

*(a)* Labour Party (UK), Gemma 2–9B

*(b)* Socialistische Partij (NL), Llama 3.1–8B-Instruct

*Figure 8.* Cosine similarity distributions between party probes and MLP value vectors after sign-inversion-based validation. Subfigure (a) shows results for the Labour Party (United Kingdom) using Gemma 2–9B, while subfigure (b) shows results for the Socialistische Partij (Netherlands) using Llama 3.1–8B-Instruct. Each distribution depicts cosine similarities $\cos(\theta_i^l)$ between the party probe weight vector $W_o$ and MLP value vectors $v_i^l$ across layers. Boxplots indicate the interquartile-range used to identify statistically extreme probe-aligned $(\cos(\theta_i^l) > 0)$ and diametric $(\cos(\theta_i^l) < 0)$ value vectors. Across both panels, a large concentration of selected value vectors appears in earlier layers, typically around one half to two thirds of the total number of layers. Only value vectors whose sign inversion decreases the median log-probability of the corresponding party token are retained for mechanistic forecasting.

Table 4. Top tokens of value probe and value vectors for Qwen3-14B model.

| MODEL | PARTY / CANDIDATE | TOP VALUE **PROBE** TOKENS | TOP VALUE **VECTOR** TOKENS |
|---|---|---|---|
| Qwen3-14B | Conservative Party of Canada | "â̄»", "ĠHyde", "ĠCOPYING", "htag", "à¨·±ä¹IJàÏ" | "icles", "ellation", "nect", "apult", "ushman" |
| Qwen3-14B | Bloc Quebecois | "translate", "Translate", "ég·", "ĠBras", "Ġfavor" | "æ³kâL½", "ulaire", "ĠFrance", "nb", "France" |
| Qwen3-14B | Liberal Party of Canada | "reta", "çtg", "æ¾¹", "æhã°°", "CM" | "raries", "itud", "RARY", "rador", "ngth" |
| Qwen3-14B | Alternative für Deutschland | "lag", "Ġrooting", "ĠAssad", "_STORE", "deps" | "uated", "uated", "rog", "ĠhÆ°à»Lng", "IrÃ¡" |
| Qwen3-14B | Sozialdemokratische Partei Deutschlands | "mine", "mpi", "ĠShapiro", "onto", "mate" | "aneously", "æLĶ", "itize", "âĵ»", "andard" |
| Qwen3-14B | Christlich Demokratische Union | "âÎ", "à½āÎ", "éÎ", "è¾¼çÎ", "ægÑæk¬" | "â̧", "â2", "ç»Lâ¥", "ç»§ç»Ñà¿ætg", "ÑĶ" |
| Qwen3-14B | Bündnis 90/Die Grünen | "âiijèµ", "elor", "æ¬", "issan", "è«Ñ" | "clÃ©", "â¼ègÎ", "é©", "fÄ¼hÎ", "æĤ" |
| Qwen3-14B | Die Linke | "âĵà®Î", "æ°âĵ", "æÌ", "ÑĶ", "âĵ»Î" | "Ġleft", "âĵ", "ĠLeft", "ĠLEFT", "left" |
| Qwen3-14B | Freie Demokratische Partei | "âÎ", "holm", "æ¥âĵâÎ", "çĵ°", "éĠĵèĵ½" | "ĠLauderdale", "ĠFridays", "(F", "çĵâeÎL", "ĵLÎ" |
| Qwen3-14B | GroenLinks-PvdA | "ainer", "ĠZam", "è�³", "âÑ©", "/std" | "presso", "ç«LᢠG", "çK", "âᵒᵒ", "âᵱ©æĶ", "âᵱ̄" |
| Qwen3-14B | Partij voor de Vrijheid | "Ġliberty", "jej", "çĵæÎ", "alytics", "â±" | "ners", "icipants", "icipant", "icipation", "ite" |
| Qwen3-14B | Volkspartij voor Vrijheid en Democratie | "âᵱ§âÎ", "éᵗÑ", "âᵢ", "çÎâÎ", "icles" | "achts", "robe", "lung", "ipeg", "atile" |
| Qwen3-14B | Socialistische Partij | "æLLâᵢÎ", "èᵢĵèhᵌ", "âlÎ", "ĐᾰÑĐµÑĝĐᵒÑi", "ĠZucker" | "æᵒµæLĵèL", "Ø§Ø©", "ç»Î", "âÎ°âᵢΰè" |
| Qwen3-14B | Democraten 66 | "â¼ĠéÎ", "ĠCOPE", "gly", "Ġnod", "âᵱâᵱ" | "ials", "ĠWithEvents", "çKLâᵒ§æG»âĠ¼", "iating", "iates" |
| Qwen3-14B | Forum voor Democratie | "ncy", "atics", "æÎçÎÎ", "çÎ", "Ent", "èÎ" | "ĠLauderdale", "ĠFridays", "(F", "çĵâeÎL", "ĵLÎ" |
| Qwen3-14B | National Party | "çÎ¥", "ĠTerms", "Ġreb", "âᵢ", "Ġnb", "ĠQatar" | "éᵢIJ", "utilus", "avigator", "omencl", "agra" |
| Qwen3-14B | Labour Party | "æ±Lâᵢ", "ĠMarxist", "RARY", "Marshal", "uds" | "Ġsocialist", "çᵌ¾âᵢ¼lâᵢ»âÎ", "éᵒâiij½", "ĠSocialist", "Ġsocialism" |
| Qwen3-14B | ACT New Zealand | "èᵌ", "âᵒâᵌ¾Ø", "çijÎ", "/about", "çᵊæÎ" | "fest", "/the", "ĠJR", "ç¬", "âᵢÎâÑ" |
| Qwen3-14B | Green Party | "ç©±âÑIJ", "(er", "éÎ¾", "èᵢè", "ĠPoz" | "ègK", "antt", "ĠGrid", "keepers", "keeper" |
| Qwen3-14B | Labour Party | "é»»", "rack", "oping", "ennie", "éᵥ" | "hound", "Øᵢ", "ĠUᵢgÛÎ", "ORK", "ĠØ§ÛØ°ÛÎ" |
| Qwen3-14B | Reform UK Party | "âᵢ", "ĠPitch", "éᵾæÎ", "ĐᵾĐ°ÑÄ", "æK" | "fest", "/the", "ĠJR", "ç¬", "âᵢÎâÑ" |
| Qwen3-14B | Conservative Party | "çᵌ", "omi", "æÎiㆰᵌ", "Closed", "ption" | "ĠCorner", "ucker", "Ġnors", "appa", "âᵢ»âᵢæGæhᵌ" |
| Qwen3-14B | Liberal Democrat Party | "éᵒ«âᵱ", "æᵒâᵢᵒᵒ", "èᵌ°æ¥", "âᵢ¼âᵱᵢh", "ØᵒÛÎ" | "raries", "itud", "RARY", "rador", "ngth" |
| Qwen3-14B | Kamala Harris | "rf", "âᵢÎ", "rv", "clip", "ĠAlta" | "Ġleft", "âᵢ", "ĠLeft", "ĠLEFT", "left" |
| Qwen3-14B | Donald Trump | "çᵌLâᵱ", "edic", "sdale", "éᵢᵌçᵢᵢ", "çᵢᵌ¾" | "çᵢâᵢ»âÑIJ", "Ġhimself", "âᵢjÎᵢâᵥiᵌ", "/she", "âᵍiâÑ" |

## B.5. Intervention in Non-Political Contexts

To complement the local necessity test in Equation (12), Figure 9 evaluates the selected value vectors in a controlled non-political setting where the persona structure is preserved but the question is unrelated to political preference. Targeted scaling of the selected vectors significantly increases the party-token logit relative to a no-scaling baseline, whereas a matched random-vector control of equal magnitude shows no significant effect. This rules out that the observed effects arise from arbitrary or redundant directions and establishes that the selected vectors functionally encode party-relevant information.

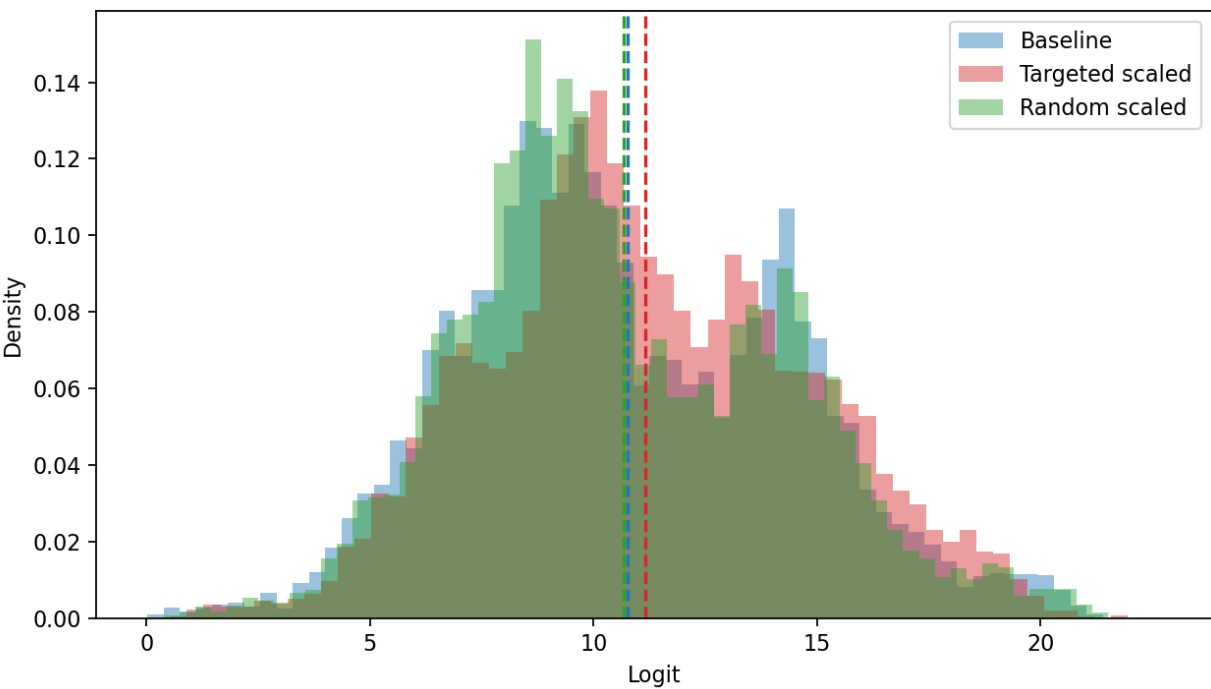

*Figure 9.* Intervention effects of selected value vectors on party-token logits in non-political contexts. Distribution of first-token logits for party tokens across $n = 1280$ controlled non-political persona–question pairs (e.g., "What ice cream flavor would I order?"), where the full persona structure is preserved but the question is unrelated to political preference. For each prompt, we compare three conditions: a no-scaling *baseline*, a targeted intervention *targeted scaled* that scales the value vectors selected via our sign-inversion procedure (Eq. 12), and a matched random intervention *random scaled* using the same number and scaling magnitude of arbitrary directions. Significance annotations report paired $t$-tests comparing each intervention condition against the baseline on identical prompts, isolating the effect of the manipulation. Targeted scaling of the selected value vectors increases the party-token logit relative to baseline across all party-level aggregates, whereas the matched random intervention control shows no significant effect. Together, these results provide evidence that the selected value vectors encode party-relevant directions and rule out that effects arise from arbitrary or redundant directions in representation space.

### B.6. In-Context Learning Baseline with Party-Position Data

A natural concern is whether the gains of mechanistic forecasting stem from access to the country-specific party-position dataset used during probe training, rather than from latent structure in the model itself. To isolate this, we construct an in-context learning (ICL) baseline that receives the *same* party-position data through the prompt rather than through probe training. For each country, we build ICL prompts by combining value-probe examples (statement, answer, comment) with the persona prompt variants used in the main estimation setup, sampling one in-context example per party to yield a balanced context, and then score all candidate parties under the corresponding LLM.

Figure 10 reports median absolute error across models, comparing the ICL baseline against our approach. The ICL baseline performs substantially better than chance, confirming that the party-position data carries useful signal, but remains worse than mechanistic forecasting for most parties. This indicates that the observed gains are not explained by access to the external position data alone, but by how latent party-relevant structure is identified and aggregated.

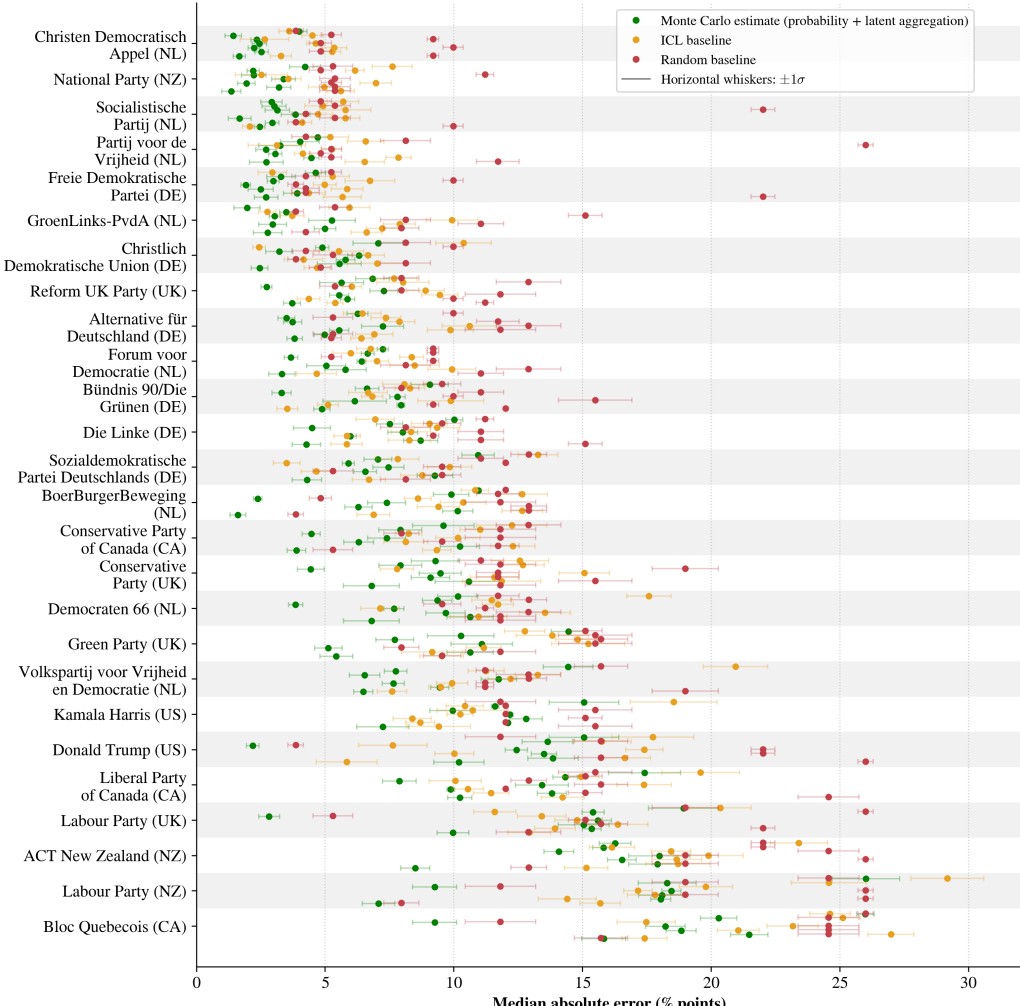

*Figure 10.* Party-level estimation error for the ICL and random baseline compared to mechanistic forecasting, following the format of Figure 3. Each point shows the median absolute error in estimating $P(\text{party} \mid \text{category})$ relative to survey benchmarks, with offsets indicating different models. The ICL baseline receives the same country-specific party-position data through the prompt that underlies our probing pipeline. Mechanistic forecasting outperforms the ICL baseline for most parties and models.

# C. Further Information on Distance Differences

Figure 11 complements the win-rate analysis in Section 5 by showing the full distribution of attribute-level distance differences $\Delta_k$ across models and countries, rather than only the proportion of attribute-level wins. This view exposes how strongly mechanistic forecasting outperforms or underperforms probability-based estimation in each setting, beyond the binary win/loss summary reported in the main text.

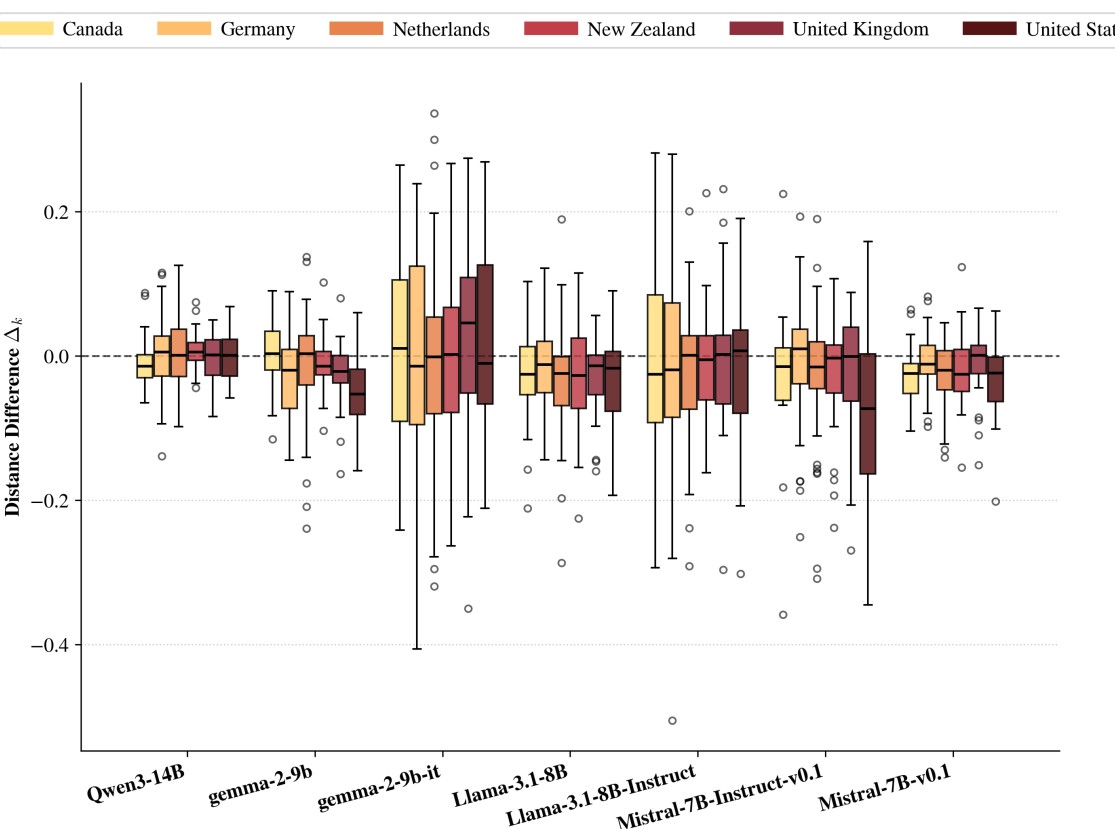

*Figure 11.* Distribution of distance differences across models and countries. Boxplots show the attribute-level distance difference $\Delta_k = D_k^{\text{prob}} - D_k^{\text{latent}}$ for each model-country pair, aggregated over persona attributes and parties. Positive values indicate cases where latent activation-based distributions are closer to survey benchmarks than probability-based estimates. The dashed horizontal line marks $\Delta_k = 0$, separating latent wins from probability wins; colors denote countries.

