# OpenReview forum: "Reading Between the Tokens: Improving Preference Predictions through Mechanistic Forecasting"
_ICML.cc/2026/Conference — ICML 2026 regular_

### Official Review · Reviewer_ayTR · 2026-03-02

**Soundness:** 2
**Presentation:** 3
**Significance:** 1
**Originality:** 3
**Overall Recommendation:** 3
**Confidence:** 3

**Summary:**

This paper explores how to improve human preference prediction, using election prediction as an example, by leveraging internal representations in LLMs. The authors argue that existing methods rely solely on model output probabilities and overlook the information present in the model’s internal latent representations. To address this, the paper uses a linear probe to identify MLP value vectors associated with political parties and computes the activation of persona prompts on these value vectors.

**Compliance With Llm Reviewing Policy:**

Affirmed.

**Final Justification:**

Thank you to the authors for the additional experiments. The new intervention results and the matched ICL baseline substantially alleviate my concerns about the validity and fairness of the method. While a few details still could be strengthened in the revision, the additional evidence is meaningfully more convincing than before.

**Key Questions For Authors:**

The questions are detailed in the Weaknesses section.

**Limitations:**

yes

**Strengths And Weaknesses:**

Strengths

1. The use of MLP value vector decomposition for identifying "party-aligned representations" is innovative.

2. The structure of the paper is clear, with a complete description of the pipeline, and the mathematical formulations are well-presented.

Weaknesses

3. The approach is essentially probing combined with feature selection, rather than true mechanistic circuit identification.

4. The paper does not provide probing selectivity analysis or random probe control.

5. The baseline comparisons are severely lacking, with only comparisons to next-token probability and latent aggregation. Key baselines are missing, including calibrated probability, logit lens aggregation, representation regression baseline, and prompt ensemble baseline.

6. There is a lack of statistical significance testing.

---

> ### Author Rebuttal · Authors · 2026-03-30
>
> Thanks for your feedback. Our goal is to understand if, when, and why latent representations provide advantages over probability-based preference prediction. We show that gains are structured: they depend on persona attributes, vary across national contexts, and are strongly moderated by output entropy. We appreciate your suggestions for additional robustness checks; we have conducted these analyses and summarize the key findings in the discussion below. The corresponding detailed results and figures are prepared and will be shared in the subsequent revision. These additional findings further substantiate our conclusions, and we invite you to reconsider your assessment in light of this evidence.
>
> **Question 1:**
> We respectfully disagree that our approach reduces to standard probing. Standard probing is purely predictive, whereas our method explicitly links representations to model computation: we decompose MLP updates into value-vector contributions (Eq. 4) and validate their functional role via counterfactual intervention (Eq. 12). This establishes that the identified directions are not merely correlational features, but components that functionally influence party-token generation. The linear probe is used only to identify a task-relevant direction; our contribution is to connect this direction to specific MLP value vectors and verify their effect on model outputs. We therefore move beyond probing toward a mechanistic characterization of how party-relevant information is encoded and used. If the method were purely probing-based, these intervention effects would not systematically hold.
>
> **Question 2:**
> Thank you for highlighting the importance of probing controls and selectivity. We add two controls: (i) label-shuffled probes and (ii) random-weight probes, both performing near chance (macro-F1: 0.33 ± 0.11 and 0.24 ± 0.15 vs. 0.99 ± 0.01), ruling out spurious alignment. We further measure probing selectivity, which is consistently large and positive across models, countries, and seeds, with the $0.5L$–$0.9L$ range (our selection) showing the most stable separation. We also add permutation-based significance testing, confirming all differences are statistically significant. Finally, regression probes perform good (0.93 ± 0.05), but remain consistently below our value probe, indicating results are less well explained by generic linear readout.
>
> **Question 3:**
> We agree and expanded the baseline comparison accordingly. Prompt ensembling is already part of our setup (10 templates per persona; see reviewer 2qyJ). In addition, we introduce representation-level baselines that use the same activations but omit our value-vector extraction and aggregation: (i) direct value-probe scoring, (ii) regression-based probes, and (iii) a random baseline. This isolates whether gains stem from latent representations in general or from our mechanistic pipeline.
> Empirically, direct probe baselines do not match our method for most parties, indicating that improvements arise from identifying and aggregating party-aligned value vectors rather than generic latent classification. In sum, we now explicitly include regression probing (see above), direct representation baselines (value + regression probe),  prompt ensembling (already included) and statistical significance testing (see below). We also clarify that logit-lens-style approaches are related but do not isolate party-specific directions.
> While calibration is relevant in principle, deterministic decoding makes temperature scaling inapplicable here. Our goal is not to replace probability-based methods, but to show that latent approaches can provide complementary signal, which we clarify in the revision.
>
> **Question 4:**
> We agree on the importance of significance testing and extended the evaluation accordingly. First, we now report the statistical significance for the value probe comparisons discussed above (reviewer 2qyJ), including comparisons against regression and control baselines. Second, for the main downstream results, we add our significance testing for the party-level estimation errors shown in Figure 3, demonstrating that the observed differences between mechanistic forecasting and probability-based estimates are statistically reliable for most models and parties across categories. Third, we include significance analysis for the improvements reported in Figure 5, focusing on the high-entropy regime, where our method is designed to provide gains.

---

> > ### Author Rebuttal · Reviewer_ayTR · 2026-04-01
> >
> > Thank you to the authors for their response. In particular, Replies 2 and 4 have addressed some of my concerns. However, regarding Replies 1 and 3, I still have several follow-up questions that I hope the authors can clarify further.
> >
> > The authors state that they validate the functional role of the selected value vectors via counterfactual intervention (Eq.~12). How does this procedure affect the final prediction accuracy? Is there any ablation study evaluating its contribution?
> >
> > Eq.12 seems to provide only an algorithmic criterion for selecting value vectors. Does it truly establish a causal relationship with the representation of a given political party, rather than merely a correlation? As currently presented, Eq.~12 appears to be only a local necessity test, which still falls short of a rigorous counterfactual intervention. Could the authors design more rigorous experiments to validate one or two of the following claims?
> >
> > A. The selected value vectors causally encode the concept of the corresponding political party itself.
> > B. They are stable causal factors in the mapping from persona to party preference.
> > C. They are not merely bypass signals that can be replaced by other redundant circuits.
> > The proposed method uses country-specific political party position data. Was the same information also used in the baseline method? If not, is this comparison fully fair? More importantly, can the authors demonstrate that the effectiveness of the method comes from hidden knowledge in the model, rather than from the additional country-specific party position data? I would recommend that the authors compare against an ICL-based baseline using the same country-specific political party position data to verify this point.
> >
> > The authors' responses to the above three questions will play a decisive role in my final rating.

---

> > > ### Author Response · Authors · 2026-04-06
> > >
> > > Thank you for the follow-up; this helps clarify the core concern.
> > >
> > > Beyond Eq. 12, we added two new intervention experiments to directly test causal influence. First, we construct a controlled dataset of non-political persona–question pairs (n=1280; e.g., “What ice cream flavor would I order?”), while keeping the full persona structure identical to the main pipeline. For each prompt, we run a paired intervention (no-scaling baseline vs. scaling) of our value-vectors and evaluate the logit/probability of the party token. Using a paired t-test (same prompt evaluated under baseline and intervention, isolating the effect of the manipulation), we find that scaling significantly increases the party-token logit/probability across all party-level aggregates on these non-political persona-question pairs, supporting (A) that the selected vectors encode party-relevant directions. \
> > > Second, we compare against a matched random-vector control (same number and scale of interventions), which shows no significant effect relative to baseline, while the targeted intervention remains significant. This addresses (C) by ruling out that effects arise from arbitrary or redundant directions. Importantly, we also observed in prior experiments that without the Eq. 12 gating (sign-inversion selection), interventions do not produce consistent or significant effects, indicating that Eq. 12 is not merely an algorithmic filter but critical for isolating functionally relevant components. We agree that isolating contribution to downstream accuracy is important and will include these ablation experiments (A+C) in the revision.
> > >
> > > **On the fairness of our baseline method**\
> > > To address this concern, we additionally implemented an ICL baseline using exactly the same country-specific party-position dataset that underlies our probing pipeline. Concretely, for each country we build ICL prompts by combining value-probe examples drawn from the election-position dataset (statement, answer, comment) with the same persona prompt variants used in the main estimation setup. For each value-probe prompt variant, we sample one in-context example per issue owner/party, yielding a balanced context over all parties, and then score all candidate parties with Llama 3.1–8B-Instruct. We currently report this baseline only for Llama 3.1–8B-Instruct, due to compute constraints during the last week, but we will extend it to all models in the revision. Using median absolute error for comparison, this ICL baseline performs substantially better than random assignment, but *remains worse than our method* for most parties, suggesting that the observed gains are not explained by access to the external party-position data alone, but by how the latent party-relevant structure is identified and aggregated.
> > >
> > > We hope these additional experiments resolved your remaining concerns and invite you to update your score based on these last experiments. Thanks!

---

### Official Review · Reviewer_aCKM · 2026-03-08

**Soundness:** 3
**Presentation:** 3
**Significance:** 2
**Originality:** 3
**Overall Recommendation:** 5
**Confidence:** 2

**Summary:**

This paper frames LLM-based social simulation as a problem of latent representation representativeness. They train probes to classify whether a political opinion belongs to a specific party and then measure the similarity of the probe weights with the activation vectors generated by different input personas to create a distribution of opinions.

**Compliance With Llm Reviewing Policy:**

Affirmed.

**Final Justification:**

Authors added more explanations about probe robustness, which i saw as the main limitation.

**Key Questions For Authors:**

1. Did you do any ablations with the layers you used for probe training? Do the results still hold if you used $l \in [0.7L, 0.9L]$?
2. How many probes did you train for each party? How much variance is there in your results between the same probe trained with different initializations?
3. Do all the probes you trained exceed 95% F1 score? This is interesting since I would expect these probes to do better on some parties more than others.
4. Did you consider a multiclass classification set up? Reading party distributions from a multiclass head seems like an important baseline to include for this discussion. If it does better than your binary probe + similarity set up, it may negate your point.

**Limitations:**

The main limitation is that this results depends on the quality of the probe and cosine distance. It is possible that there are other ways to use activations that achieve very good predictions of the actual population distribution. But I don’t think this is a critical flaw, proving that something doesn’t exist is much harder than proving it does.

**Strengths And Weaknesses:**

Originality: This paper has a really interesting motivation. It addresses the limitations of LLMs for simulating public opinion data by demonstrating that interpreting model internals also does not produce enough relevant information. This paper presents an interesting direction of using models to prove what models might not be able to do well.

Soundness: this paper takes steps to ensure methodology is sound such as testing the probes validity. They also run experiments in different languages for prompt schemes for different countries which is necessary. For example, the authors confirm a high F1 accuracy for their political party probes.

Significance: I think it is unsurprising that using probes gives better estimates than just using model probabilities. But it is nice to have this paper show it in a systematic investigation.

Presentation: I think in general the paper is well written and clear. I think some of the take-aways are less clear. For example, Figure 2 shows a mixed bag of win rates, it's hard to draw any distinct conclusions. Perhaps an average across all countries' bars would improve the presentation.

---

> ### Author Rebuttal · Authors · 2026-03-30
>
> Thank you for your valuable feedback. We address significance, the weaknesses, and questions next.
>
> **Significance:**
> Thank you for this comment. We respectfully disagree that our results are unsurprising. Prior work on hidden knowledge [1,2] shows advantages of internal representations mainly for binary factual correctness tasks. In contrast, we study population-level, distributional preference prediction via Monte Carlo aggregation over personas, which is substantially more complex and, to our knowledge, has not been examined from a mechanistic perspective. More importantly, our contribution is not simply that probes can outperform output probabilities, but when and why this occurs. We show that gains are structured: they depend on persona attribute type, vary across national contexts, and are moderated by output entropy. In particular, latent aggregation is most beneficial in high-entropy settings where probabilities are diffuse, while probability-based methods remain competitive in low-entropy regimes. We therefore interpret our results not as a limitation of LLMs, but as evidence that latent representations provide a complementary signal when surface outputs are unreliable. This shifts the focus from whether LLMs can be used for preference prediction to how they can be used effectively, which we view as the key contribution.
>
> **Question 1:** We agree that layer selection is an important hyperparameter. We now include a systematic ablation over relative layer windows, evaluating probe performance and selectivity across different depth ranges. We find that mid-to-late layers consistently yield the strongest and most stable signal. In particular, the range $0.5L$–$0.9L$ (our choice) exhibits the highest lower whisker in the selectivity-gap distribution across seeds, models and countries, indicating robust separation between real probes and label-shuffled controls. These results support our choice of $0.5L$–$0.9L$ as a stable default, while confirming that layer range should be treated as a tunable hyperparameter.
>
> **Question 2:**
> We train one probe per party. To assess robustness, we additionally computed results across multiple random initializations (seed: 0-9). We find low variance for our probe (macro-F1: $0.99 \pm 0.01$), indicating highly stable performance across runs. In contrast, control conditions (e.g., macro-F1 scores random weights: $0.24 \pm 0.15$, shuffled labels: $0.33 \pm 0.13$) exhibit substantially higher variance, as expected under weak or absent signal.
>
> **Question 3:**
> All probes achieve high performance, with an overall macro-F1 of $0.99 \pm 0.01$ for our probe. While performance is consistently strong across parties, we observe modest variation between them, which we attribute to differences in data distribution (e.g., class imbalance, party distinctiveness, and overlap in issue positions). Importantly, all parties achieve high F1 scores significantly above control baselines, indicating that the probe reliably captures party-associated structure across the full set of classes.
>
> **Question 4:**
> We agree that a multiclass head is a meaningful baseline. Importantly, however, it optimizes a different objective (joint discrimination across parties) than our setup, which is designed to recover party-specific directions for mechanistic analysis. A direct and fair comparison is to train a multiclass linear head and use each class-specific weight vector analogously to our current probe direction, preserving the downstream cosine-based extraction. We consider this a valuable extension and will discuss it as an outlook in our revision.
>
> **Limitations:**
> We agree that results depend on probe quality. To ensure robustness, we conduct extensive validation: (a) probes generalize well on held-out data (F1 > 96%) and yield semantically meaningful features, as confirmed by projecting value vectors into vocabulary space (Table 3); and (b) identified value vectors are validated via sign-inversion tests (Eq. 12), showing that reversing their contribution decreases the log-probability of the corresponding party token on held-out data. We view these as strong checks on probe validity. We also agree that other model components contribute to final outputs. We therefore position our method not as a definitive solution, but as a proof of concept that latent preference signals exist and can be exploited. Our entropy-based gating criterion is a first step toward identifying when such signals are most reliable, leaving room for future work on alternative aggregation strategies.
>
> [1] Gekhman et al. Inside-out: Hidden factual knowledge in llms.2025
>
> [2] Orgad, H. et al. Llms know more than they show: On the intrinsic representation of llm hallucinations.2024

---

> > ### Author Rebuttal · Reviewer_aCKM · 2026-04-03
> >
> > Thank you to the authors for detailed responses.

---

### Official Review · Reviewer_2qyJ · 2026-03-11

**Soundness:** 3
**Presentation:** 4
**Significance:** 3
**Originality:** 4
**Overall Recommendation:** 5
**Confidence:** 3

**Summary:**

This paper proposes mechanistic forecasting. Instead of relying on token distributions, they look at the latent representations of an LLM to more accurately predict preference distributions that reflect survey data. They show that LLMs contain party-level political information that can be used via personas. They also propose a method for deciding when their method should be preferred over probability-based prediction.

**Compliance With Llm Reviewing Policy:**

Affirmed.

**Final Justification:**

I maintain that I think this is a solid contribution. The authors helped address my questions. This makes me more confident in my score but I already said it should be accepted so I did not change the score.

**Key Questions For Authors:**

Also see weaknesses related to prompts and party token.

### Minor

Line 221 — Diametrically contributions?

Shouldn’t the removal of the diametric vectors increase the log-prob? I assume you do this for both sets but it's not the same direction for both.

This also reminds me of the of the earlier applications of e.g. ELMo that used learned task weights to reweight hidden layer representations. Did you try other hyperparameters for the layers? 0.5-0.9L seems reasonable but I’m just curious if you think we should treat those as start and end depth proportion hyperparameters.

**Limitations:**

yes

**Strengths And Weaknesses:**

### Strengths

This is an interesting and well written paper. It discusses how probes are constructed with MLPs to better understand party-aligned encodings in intermediate layers. The steps are well justified and described cleanly. The comparison with survey data is interesting and the analysis of each aspect of demographics is insightful. Seems like a useful extension to current mechanistic interpretation methods and I agree with the authors that I think it could be interesting to extend this beyond party-level forecasting and much more broadly.

### Weaknesses

What kind of methodology was used to write the prompts? I see a few in the appendix but the text mentions that there are more but not how the template was written. There is already a lot of information about the prompts but this is an area where I am often thinking about the prompt as a hyperparameter and I don’t have as clear of an idea when I’ve explored the space enough. Additional thoughts on this are appreciated.

I’m unclear about the party token concept. Is it literally just the token for the party name? Or is there more to this set of tokens? For probably too long, I thought it meant a token

---

> ### Author Rebuttal · Authors · 2026-03-30
>
> Thanks for your thorough feedback! We are happy to answer your remaining questions below:
>
> **Weakness 1 methodology for prompts:**
> Constructing the prompts requires two key decisions: which variables to include for voting predictions, and how to embed those variables in a template. For the first decision, we draw on existing social science literature that theoretically and empirically identifies which persona attributes are most predictive of voting outcomes. For the second, we use the following scheme: Starting from the prompt structure in [1], we created 5 paraphrases—each expressing the same content with different wording. For each of these 5 paraphrases, we then created a second version with the sentence order rearranged, yielding 10 prompt variations in total. All 10 variations encode the same persona, differing only in wording and sentence order. We did not find any research that gives answers as to how many variations one should try exactly; so you are right, this is an open problem. However, using 10 prompt variations is already more than what other studies use in this space, which usually do not account for prompt rephrasing-based robustness or minimally by e.g. shifting from first-person to second-person point of view (compare e.g. [1], [2], [3])
>
> **Weakness 2 party token concept:**
> Thanks for pointing out this ambiguity. By “party token,” we do not refer to a single token in all cases. Instead, we define a set of tokens associated with each party, constructed to robustly capture how the party is represented in the model’s vocabulary. In practice, party names are often split into multiple subtokens by the tokenizer; therefore, for parties whose names span multiple tokens, we aggregate over the full token sequence rather than relying on any single subtoken. Concretely, probabilities and intervention effects are computed over the complete sequence corresponding to the party name (e.g., summing log-probabilities across tokens), ensuring that the representation reflects the full lexical realization rather than artifacts of tokenization.
> In addition, we extend this set beyond the canonical party name to include common alternative surface forms, such as acronyms and widely used abbreviations, as well as frequent variants observed in our datasets. This design choice is consistent with how tokenization in LLMs can fragment or vary representations depending on context, making it important to consider multiple lexical realizations rather than a single token. The resulting token set is thus a small, semantically coherent group of tokens that jointly represent a party. We will clarify this definition in the revision (also aligned with the implementation details already provided in the config files of our anonymous code repository).
>
> **Minor point 1 wording:**
> Thanks, we changed to “diametric contributions”
>
> **Minor point 2 diametric vector:**
> Yes, this is correct: diametric vectors are defined by negative cosine similarity to the probe direction, meaning they are anti-aligned and contribute negative evidence for the target party; as a result, removing or down-weighting them removes this opposing signal and therefore increases the target party log-probability (in contrast to positively aligned value vectors, whose removal would decrease it), and we will clarify this sign asymmetry more explicitly in the paper.
>
> **Minor point 3 ELMo:**
> We agree that the layer range should be viewed as a hyperparameter, and we thank the reviewer for highlighting this point. In addition to preliminary experiments, we now include a dedicated analysis in the paper that systematically evaluates different relative layer windows. Specifically, we measure the selectivity gap across layer ranges. This analysis shows that mid-to-late layers yield the strongest and most stable signal. In particular, the range \(0.5L--0.9L\) (ours) exhibits the highest lower whisker of the selectivity-gap distribution across seeds, indicating the most consistently positive separation between the real probe and the label-shuffled control. This robustness (not just peak performance) motivated our choice of this range as the default. We emphasize that we do not claim this range is universally optimal. Rather, our findings suggest that informative signals are concentrated in intermediate-to-late layers, and the optimal range should be selected empirically depending on the model and task. This is conceptually aligned with prior work such as ELMo, where layer contributions are treated as learnable or tunable. We will clarify this empirical justification and explicitly reference the new analysis in the revision.
>
> [1] von der Heyde et al. United in Diversity? Contextual Biases in LLM-Based Predictions of the 2024 European Parliament Elections, 2024
>
> [2] Argyle, Lisa P., et al. "Out of one, many: Using language models to simulate human samples." 2023
>
> [3] Bisbee et al. "Synthetic replacements for human survey data? The perils of large language models." 2024

---

> > ### Author Rebuttal · Reviewer_2qyJ · 2026-04-01
> >
> > I understand your reasoning for the prompting but this is not in the paper. This needs substantial discussion. Is it just paraphrase and reordering? Is this done automatically? It may seem like small details but I think they are quite important.
> >
> > Is there any length bias in the party name probabilities? How is the full set created? Again, may seem like minor details, but I cannot reproduce this if I do not know the set of "common alternative surface forms, such as acronyms and widely used abbreviations, as well as frequent variants observed in our datasets".

---

> > > ### Author Response · Authors · 2026-04-06
> > >
> > > Thanks again for your feedback! Here, our responses to your follow-up questions:
> > >
> > > **Question 1:**
> > > We will add a detailed explanation of how the prompt variations were created in the paper. We did both rephrase and reorder done in a semi-automatic way. That is, we first created the ten variations for one country (Germany) by hand (to ensure that the variations are good; LLM-based rephrases were not satisfactory). Then we translated these 10 variations for the other countries automatically and checked for correctness, including the consultation of native speakers.
> > > All prompts for all countries are visible in the supplementary materials and in the code that we released for reproducibility but we agree with you that we should add this discussion much more prominent in the main body of the paper. We will do so for the revision. We will also call for the necessity of a much more rigorous framework for these prompt variation strategies. We absolutely agree with you on that.
> > >
> > > **Question 2:**
> > > We also agree that the current description of the party-token setup is not sufficiently explicit for reproducibility and will make this substantially clearer in the revision. In the implementation, the party representation is defined through an explicit, fixed set of surface forms per country and party/candidate, specified in the configuration files. This set includes the canonical party name together with common abbreviations and widely used variants from the underlying election material (e.g., for Germany, Bündnis 90/Die Grünen, Die Grünen, GRÜNE, BÜNDNIS 90/DIE GRÜNEN, …). We will add the exact construction procedure and the complete lists to the paper/appendix so that the setup is fully reproducible. We also agree that length bias is an important concern: multi-token names can otherwise be disadvantaged relative to shorter acronyms, which is precisely why we use a fixed surface-form set for the dataset instead of relying on a single realization only. In the revision, we will therefore make this explicit. Thanks for pointing this out!

---

### Official Review · Reviewer_wcaQ · 2026-03-19

**Soundness:** 2
**Presentation:** 3
**Significance:** 3
**Originality:** 3
**Overall Recommendation:** 4
**Confidence:** 3

**Summary:**

This paper proposed an argument that LLMs already encode human preferences, but LLM outputs can not reliably reflect them. To demonstrate this, the authors designed a mechanistic forecasting method that identifies party-aligned value vectors in MLPs of LLMs, measures the strength of activations conditioning on personas, and aggregates across personas to produce a preference distribution. The results show that the preference distribution obtained by mechanistic forecasting is more aligned with the real distribution in real-world surveys than the distribution obtained by output probabilities. The result indicates that latent features are more reliable signals when measuring preference, especially when the LLM is uncertain (when the entropy is high).

**Compliance With Llm Reviewing Policy:**

Affirmed.

**Final Justification:**

The authors explained more about the simplified assumptions, which addressed most of the concerns, though having experiments on bigger models or reasoning models would still be helpful.

**Key Questions For Authors:**

Does the conclusion still hold with larger-scale LLMs (e.g., with a size of 72B), or reasoning LLMs?

**Limitations:**

yes

**Strengths And Weaknesses:**

Strength:
1. The proposed method, mechanistic forecasting, is novel. The effectiveness is also well-supported by the large-scale empirical results.
2. The analysis is comprehensive. The entropy-based conclusion offers good inspiration for future works in this direction.

Weakness:

The assumptions in the proposed method are too strong. As the authors admit, the entire analysis is built on the local first-order approximation of MLP to token logits; however, repeated operations in transformers consist of MLP, layernorm, attention, etc., which are all nonlinear.

What’s more, the linear probe can be an oversimplified assumption. Even if previous works show that semantic features are often strong from intermediate representations (so linearly decodable), it does not guarantee that such information is used linearly in the model.  A stronger causal evidence to prove that the value vector really carries party-related computation is helpful (e.g., intervene on the vectors and show that the model's behavior towards party preference changes in the predicted way).

---

> ### Author Rebuttal · Authors · 2026-03-30
>
> Thank you for the thorough feedback and for acknowledging the novelty, scale and usefulness of our work! We address the raised weaknesses next.
>
> **Weakness 1: First-order approximation of MLP contributions:**
> We agree that the first-order approximation is a simplification, but our core results do not depend on it being exact. It is used only as a heuristic to identify candidate value vectors. Crucially, we validate these vectors via sign-inversion (Eq. 12), directly testing whether reversing their contribution decreases the log-probability of the corresponding party token on held-out data. This provides a model-agnostic empirical check that does not rely on linearity or additivity across layers, and only validated vectors are retained. Moreover, using first-order decompositions as local analytical tools, rather than exact models of the model’s full nonlinear computation, is standard practice in mechanistic interpretability (cf. Sections 4.1–4.2). Our contribution is therefore not the approximation itself, but showing that directions identified in this way yield intervention-validated improvements at scale.
>
> **Weakness 2: Linear probe as an oversimplified assumption:**
> We appreciate this important point. We fully agree that linear decodability does not guarantee that the model uses these features linearly. Our goal, however, is not to claim strict linear usage, but to show that latent representations contain structured, exploitable signals that can be reliably leveraged for preference estimation. To address this concern more directly, we provide three pieces of evidence. First, our sign-inversion test (Eq. 12) constitutes a form of causal intervention: we counterfactually reverse the contribution of individual value vectors and measure the resulting change in party token probabilities. As described in Section 4.2, value vectors are only retained if their removal decreases the log-probability of the corresponding party token on held-out data, demonstrating that these directions have a direct functional influence on model outputs rather than reflecting mere correlations. Second, to rule out that results arise from arbitrary linear projections, we additionally evaluated control conditions using label-shuffled probes (F1 ≈ 0.33) and random-weight probes (F1 ≈ 0.24), both of which perform near chance, indicating that probe effectiveness depends on meaningful alignment between representations and labels. Third, following also reviewer ayTR’s concern, we introduce an additional regression baseline (added in the revision) trained on the same activations. While this baseline achieves strong performance (0.93 ± 0.05 macro-F1), it is consistently outperformed by our value probe (0.99 ± 0.01; significantly better with p<0,001), showing that recovering a generic linear signal is less sufficient and that the probe learns a more discriminative, task-aligned direction. Taken together, these results show that our method does not rely on trivial linear decodability, but identifies directions that functionally influence model outputs. Finally, the observed, systematic improvements over output probabilities across 7 models, 6 countries, and 24M+ configurations are difficult to reconcile with purely spurious correlations. The goal of our paper is to show that we can effectively leverage latent information and we succeed in doing so. Figures of these additional analyses regarding probing baselines are included in the revision.
>
> **Question 1 bigger/reasoning models**:
> We would like to note that our results already demonstrate consistent effects across diverse architectures, countries, and 24M+ configurations, suggesting robustness of the underlying phenomenon. We agree that extending to larger or reasoning models is valuable future work. While this discussion period is too short to deliver results, we commit to conducting an experiment with a larger model and a reasoning model for the camera-ready version.

---

> > ### Author Rebuttal · Reviewer_wcaQ · 2026-04-03
> >
> > Thanks for your response. Most of my concerns have been addressed, so that I will raise my score.
> >
> > However, it would be a good (and necessary) experiment to validate whether all your conclusions (including your simplified assumptions) still hold for a model with more parameters, or for a reasoning model with a different training stage (RLVR).  That will make the paper more solid.

---

> > > ### Author Response · Authors · 2026-04-06
> > >
> > > We are glad that we could resolve your concerns! We will include such experiments for larger-scale/reasoning models in the camera-ready version.

---

### Decision · Program_Chairs · 2026-04-30

**Decision:**

Accept (regular)

**Comment:**

The paper studies preference prediction using model internals rather than relying solely on output distributions. In particular, it identifies party-aligned value vectors within the MLP layers of LLMs, measures activation strengths conditioned on different personas, and aggregates these signals to derive a preference distribution. Empirically, this mechanistic forecasting approach outperforms methods based on output distributions.

All reviewers acknowledged the novelty, soundness, and clarity of the work. They raised concerns about the method’s strong assumptions, its sensitivity, and the rigor of the empirical evaluation. The authors have addressed most of these concerns in the rebuttal. We therefore recommend acceptance to ICML and encourage the authors to incorporate these clarifications in the camera-ready version.